# Insights on the Role of Accurate State Estimation in Coupled Model Parameter Estimation by a Conceptual Climate Model Study

Xiaolin Yu[1], Shaoqing Zhang[1,2*], Xiaopei Lin[1,2], Mingkui Li[1]

[1]Physical Oceanography Laboratory of OUC, and Qingdao Collaborative Innovation Center of Marine Science and Technology, Qingdao, 266001, China
[2]Function Laboratory for Ocean Dynamics and Climate, Qingdao National Laboratory for Marine Science and Technology, Qingdao, 266001, China

*Correspondence to*: Shaoqing Zhang (szhang@ouc.edu.cn)

**Abstract.** The uncertainties in values of coupled model parameters are an important source of model bias that causes model climate drift. The values can be calibrated by a parameter estimation procedure that projects observational information onto model parameters. The signal-to-noise ratio of error covariance between the model state and parameter being estimated directly determinates whether the parameter estimation succeeds or not. With a conceptual climate model that couples the stochastic atmosphere and slow varying ocean, this study examines the sensitivity of state-parameter covariance on the accuracy of estimated model states in different model components of a coupled system. Due to the interaction of multiple time scales, the fast varying "atmosphere" with a chaotic nature is the major source of the inaccuracy of estimated state-parameter covariance. Thus enhancing the estimation accuracy of atmospheric states is very important for the success of coupled model parameter estimation, especially for the parameters in the air-sea interaction processes. The impact of chaotic-to-periodic ratio in state variability on parameter estimation is also discussed. This simple model study provides a guideline when real observations are used to optimize model parameters in a coupled general circulation model for improving climate analysis and predictions.

# 1 Introduction

Nowadays, a coupled atmosphere-ocean general circulation model is widely used as a common tool in climate research and related applications. However, due to the approximation nature of model numeric schemes and physical parameterization, a model always has errors. In particular, one traditionally determines the values of model parameters by experience or a trial procedure which heuristically provides a reasonable estimate but usually not optimal for the coupled model. Recently, with the aid of information estimation (filtering) theory (e.g. Jazwinski, 1970), researches on optimization of coupled model parameters based on instantaneous observational information have grown quickly (e.g. Wu et al., 2013; Liu et al., 2014; Liu et al., 2014; Li et al., 2016). Traditional data assimilation that only uses observations to estimate model states (i.e. state estimation) becomes both state estimation (SE) and parameter estimation (also called optimization) (PE) with observations. Such a PE process can be implemented through a variational (adjoint) method (e.g. Stammer, 2005; Liu et al., 2012), or an ensemble Kalman filter (e.g. Zhang et al., 2012), or even a direct Bayesian approach (e.g. Jackson et al., 2004).

In the previous study with a conceptual coupled model, Zhang et al. (2012) pointed out that an important aspect of successful coupled model parameter optimization is that the coupled model states must be sufficiently constrained by observations first. This is because multiple sources of uncertainties exist in a coupled system consisting of different time scale media. If the part of uncertainties in model states, which is not correlated with parameter errors, has not been sufficiently constrained yet, the covariance between the model states and parameters being estimated is noisy (e.g. Dee & Silva, 1998; Dee, 2005; Annan et al., 2005). Without direct observational information, the noise in state-parameter covariance that is the key quantity to project observed state information onto the parameter, can bring the estimated parameter toward an erroneous value (Zhang et al., 2011b). This is a general understanding about coupled model parameter estimation. However, since multiple media of the climate system have different time scale variability and different quality of observations so as to have different contributions to the uncertainty of state-parameter covariance, an outstanding question is: what is the impact of SE accuracy in different media on coupled model PE? Given the extreme importance of state-parameter covariance for PE, a clear answer for this question must further our understanding on coupled model parameter estimation.

To answer this question, this study uses a simple coupled model to examine the influence of observation-constrained states in each medium on PE for different parameters in different media thoroughly. The model conceptually describes the interactions of 3 typical time scales of the climate system – chaotic (synoptic) atmosphere, seasonal-interannual upper ocean and decadal deep ocean. A twin experiment framework is used throughout the whole study.

The paper is organized as follows. After introduction, section 2 gives methodology, including brief descriptions on the simple coupled model, filtering algorithm and twin experiment framework. Section 3 first presents the results of various PE experiments with different partial SE settings, and then analyzes the conditions for successful PE with partial SE. Finally, summary and discussions are given in section 4.

## 2 Methodology

### 2.1 The model

To clearly address the issue posed in the introduction, this study employs the simple pycnocline prediction model developed by Zhang (2011ab). This conceptual coupled model is based on the Lorenz's 3-variable chaotic model (Lorenz, 1963) that is coupled a slab ocean variable (Zhang et al., 2012) interacting with a pycnocline predictive model (Gnanadesikan, 1999). For the problem that is concerned, this conceptual coupled model shares the fundamental features with a coupled general circulation model (CGCM) (see Zhang, 2011a; Han et al., 2013). The model development can be traced in Zhang (2011ab) and Zhang et al. (2012) in details. Here, we only comment on major points that are relevant to this study. The model includes 5 equations:

$$\dot{x}_1 = -a_1 x_1 + a_1 x_2$$
$$\dot{x}_2 = -x_1 x_3 + (1 + c_1 w) a_2 x_1 - x_2$$
$$\dot{x}_3 = x_1 x_2 - b x_3 \tag{1}$$
$$O_m \dot{w} = c_2 x_2 + c_3 \eta + c_4 w \eta - O_d w + S_m + S_s \cos(2\pi t / S_{pd})$$
$$\Gamma \dot{\eta} = c_5 w + c_6 w \eta - O_d \eta$$

The first 3 equations (Lorenz's 3-variable chaotic model) represent the dynamics of "atmosphere." The last 2 equations respectively represent the dynamics of the slab upper ocean and the pycnocline depth variation of deep ocean. There are 5 variables in the model. $x_1$, $x_2$ and $x_3$ are the fast-varying variables of the atmosphere with the parameters $a_1$, $a_2$, $b$ set as 9.95, 28, and 8/3, that sustain the chaotic nature of the atmosphere. $w$ and $\eta$ are the low-frequency variables of the ocean. Equation (1) tells that the ocean in this system is driven by two kinds of forcings: the chaotic $x_2$ from the Lorenz equations and the periodic cosine function term serving as the external forcing of the system. The coupling parameter $c_2$, which interacts with the chaotic forcing $x_2$, is set as 1. $O_d$ is damping coefficient. In this simple model, the damping coefficient is set to be identical for the upper ocean and deep ocean as 1. Values of other parameters such as $c_1$, $c_3$, $c_4$, $S_m$, $S_s$, $S_{pd}$, $O_m$, $\Gamma$, $c_5$, $c_6$ are set as $10^{-1}$, $10^{-2}$, $10^{-2}$, 10, 1, 10, 10, $10^2$, 1, $10^{-3}$ (the justification can be found in the literature cited before). The upper ocean is slower than the atmosphere due to great heat capacity of water. The deep ocean is slower than the upper ocean due to lack of mixing. The parameter $O_m$ ($\Gamma$) that represents the heat capacity of upper (deep) ocean, combining with the damping coefficient $O_d$, defines the fluid characteristic time scale. For example, the ratio $O_d/O_m$ of $10^{-1}$ ($O_d = 1$, $O_m = 10$) defines the characteristic time scale of $w$ being 10 times of that of $x_2$. It is important to mention that, with these parameter settings, the chaotic atmospheric forcing is stronger than the periodical forcing in the "ocean" equation. The "ocean" feeds back to the "atmosphere" in the low-frequency band, therefore in this coupled model, the uncertainty caused by chaotic "atmosphere" spreads to whole range of resolved periods for both the "atmosphere" and the "ocean". From Eq. (1), it can be seen that the parameter $a_2$ has a direct influence on the variation of the state variable $x_2$. And the parameter $c_2$ has a direct influence on the variation of the state variable $w$. The estimation of these two parameters will be used later to interpret the relation between

the accuracy of SE and successfulness of PE. Although very simple, this low-order (limited-size) conceptual model mimics very fundamental natures of interactions of three typical time scales in the real world: synoptic (chaotic) atmosphere, seasonal-interannual upper tropical oceans and decadal/multi-decadal deep ocean (Zhang 2011b). The boundary condition is a predefined seasonally-varying solar radiation $S(t)=S_m+S_s\cos(2\pi t/S_{pd})$. The state variable $w$ mimics the surface temperature of the ocean and the $x_2$ mimics the surface wind of the atmosphere. Here we may mimic some parameterization of CGCM using the relation of parameters and model variables ($c_2$ and $x_2$ analogous to the drag coefficient $c_d$ and wind for the stress on ocean, for instance).

## 2.2 Filtering scheme

The filtering method used in this study is the ensemble adjustment Kalman filter (EAKF, Anderson, 2001). The EAKF algorithm shares all theoretical derivation of ensemble Kalman filter (EnKF, e.g. Evensen, 1994; Houtekamer et al., 1998) that combines observational probability distribution function (PDF) with model PDF but under an adjustment idea. After the first version (Anderson, 2001), the EAKF algorithm had improved its implementation as a sequential local least squares filter (Anderson, 2003). The EAKF is a member of ensemble square root filters (Tippett et al., 2003), taking the advantage of ensemble Kalman filter without perturbing on the observation (Whitaker and Hamill, 2002). While the detailed and exhausted mathematical derivations can be referred to the literature aforementioned and others (e.g. Zhang and Anderson 2003), here we mainly comment on the computational implementation with a two-step procedure (Anderson, 2003; Zhang et al., 2007) that is relevant to this study. The first step uses two Gaussian convolution to derive the observational increment at the observational location as:

$$\Delta y_i^o = \underbrace{\frac{\frac{1}{(\sigma^p)^2}\overline{y}^p + \frac{1}{(\sigma^o)^2}y^o}{\frac{1}{(\sigma^p)^2}+\frac{1}{(\sigma^o)^2}}}_{\text{adjusted ensemble mean}} + \underbrace{\frac{\Delta y_i^{\,p}}{\sqrt{1+(\frac{\sigma^p}{\sigma^o})^2}}-y_i^{\,p}}_{\text{adjusted ensemble spread}} \qquad, \ (i{=}1{\sim}N, \text{ N is the ensemble size}) \qquad (2)$$

Where $y$ represents the observable state variable and $\sigma$ is its error standard deviation. A superscript $p$ always denotes the prior quantity estimated by model, and $o$ denotes observational quantity. An over-bar denotes the ensemble mean.

The second step regresses the observational increment onto the related model states or parameters by the model ensemble-evaluated covariance as:

$$\Delta p_i^u = \frac{\text{cov}(\Delta p, \Delta y)}{\text{std}(\Delta y)^2} \Delta y_i^o, \quad \text{(i=1~N, N is the ensemble size)} \tag{3}$$

The linear regression Eq. (3) is built with the help of the 20-member ensemble. $\Delta p_i^u$ is the adjusted state (parameter) increment given the observational increment $\Delta y_i^o$. cov$(\Delta p, \Delta y)$ is the error covariance computed between the ensembles of the model variable at the model grid and at the observational location (for SE) or between the ensembles of the state variable and perturbed parameter being estimated (for PE). std$(\Delta y)$ is the standard deviation of the ensemble of state variable at the observational location. For example, when using $x_2$ to estimate $c_2$, on each estimating step, the ensemble of $x_2$ and the ensemble of $c_2$ are used to calculate the ratio of cov/std$^2$ and adjust $c_2$ toward a better value that minimizes the errors of model states from the observations. While such a sequential implementation provides much computational convenience for data assimilation, the EAKF maintains the nonlinearity of background flows as much as possible (Zhang and Anderson, 2003; Zhang et al., 2007). It's worth mention that like usual EnKFs or variational methods without a model error compensation term, the EAKF has the disadvantage on dealing with model errors.

Some other relevant aspects of the method are also commented here. Same as in Zhang and Anderson (2003), based on the trade-off between cost and assimilation quality, after a series of sensitivity tests on ensemble sizes of 10, 20~100, no significant difference on the quality of standard assimilation is found when the ensemble size is greater than 20. Thus a practical ensemble size of 20 is chosen as a basic experiment setting. We will examine the sensitivity of major conclusions of the addressed problem in this study to the ensemble size in related places later. Although the intervals of the atmosphere and ocean observations are different in the real world, for convenience of comparison, we set a uniform update interval for SE (in the atmosphere and ocean) and PE as 5 time steps as the basic setting in this study (we will also discuss the influence of update intervals in related places later). The inflation method must be included in the EAKF PE. Considering that the inflated parameter ensemble will influence on state variables, no inflation is applied to the model state ensemble. The PE inflation scheme follows Zhang (2011b): when the std (spread) of the parameter ensemble is below some limit (40% of the initial spread), a factor is applied to inflate the parameter ensemble spread to this value. During this process, the ensemble structure of parameter remains unchanged. In addition, to avoid the uncertainty and complexity of evaluating cross covariance between media that have too different characteristic time scales (Han et al., 2013), in the SE of this study, we only allow $x_2$ observations impact on all $x$ variables, and w ($\eta$) observations impact on w ($\eta$) itself, while the PE could use different medium observations.

## 2.3 Twin experiment setup

Twin experiments are set to test the relation between coupled SE and PE. The model with the standard parameter values described in section 2.1 is running $10^3$ time units (TUs) after a spin-up of $10^3$ TUs ($2 \times 10^3$ TUs in total). Here a TU is a dimensionless time unit as defined in Lorenz (1963), roughly referring to the time scale of atmosphere going through from an attractive lob to the other (1 TU equals 100 steps of the model integrations with a $\Delta t$ of 0.01). The output of last $10^3$ TUs

is then used as the "truth" to produce "observations." The "observations" are sampled as the "truth" values superimposed by a white noise with an observational interval (5 time steps in this case). To simplify, we sample the "atmosphere" observations by $x_2$ and "ocean" observations by $w$ as the basic experiment setting. The standard deviation of "observational" errors, (from the in-situ instruments, for example), are 2 for $x_2$ and 0.2 for $w$ in our cases. Note, what we describe here is a
kind of observing system simulation experiments (OSSEs, e.g. Tong and Xue, 2008; Jung et al., 2010). The assimilation model control is an ensemble of integrations for each test case with the perturbed parameters on an erroneously-set parameter value (will be described later). The initial conditions of the ensemble for assimilation are taken from the end of $10^3$ TUs spin-up (the different members in the ensemble are all resulted consequence from the parameter perturbation).

The first set of PE experiments is to study the parameters on "air-sea" interaction. To do that, we use two parameters – $a_2$
in the atmosphere equation and $c_2$ in the ocean equation to perform PE experiments. We first conduct two PE cases with full SE – both $x$ and $w$ are constrained by their observations. Then we conduct 8 PE cases with partial SE – only some medium is constrained by its observations as listed in Table 1. Through thoroughly analyzing these PE cases with partial SE, which have different SE accuracy, we are able to detect the influence of the SE accuracy in different medium on coupled model PE. After the first set PE experiments, we also conduct a second set of PE experiments to examine the influence of state
estimation accuracy on "deep ocean" parameter $c_6$ using $\eta$ observations (the observational error is set as 0.1).

In all PE cases, the initial value of the parameter to be estimated is deliberately set biased from the "truth" (referring to the standard parameter values described in section 2.1). To maintain the chaotic nature of the Lorenz equation, parameter values are required being within a certain range. This is a constraint for the biased amount of the initial values of a parameter. Based on some sensitivity studies, the chaotic performance is more vulnerable to the change of the atmospheric parameter $a_2$
than to the change of the oceanic parameters. Therefore we set the ensemble initial values of $a_2$ as a Gaussian distribution $N(30, 1)$ (30 as the mean and 1 as the standard deviation), and the spread is enough for the model ensemble uncertainty. The ensemble initial values of $c_2$ are set as $N(0.8, 0.5)$ (restricted to be positive definite). If PE is successful, then the ensemble mean value of $a_2$ ($c_2$) should converge to 28 (1). In all PE experiments, the PE is activated after 80 TUs of SE that constrains the model states close to the observations so as to enhance the parameter-state covariance for the coupled PE (Zhang et al.,
2012). The delayed time scale of PE from SE will be discussed later .

**3 Impact of SE accuracy on coupled model PE**

With the method and experiment settings described in section 2, we test different PE performances under different SE settings. Generally, with a full SE (all the atmospheric $x_{1,2,3}$ and oceanic $w$ states are estimated with the "observations" that sample the "truth"), the PE is steady and successful, no matter what observations are used to estimate which parameter. For
example, the result of using observations of $w$ (in the ocean)  to estimate $a_2$ (parameter in the atmosphere) with all simulated $x_{1,2,3}$ and $w$ being estimated by $x_2$ and $w$ observations is shown in Fig. 1a. We can see the ensemble of $a_2$ successfully converges to the "truth" from the initial biased values around 30. However, if only a part of observations (only one medium

observations) is used in SE, then the PE succeeds in some cases but fails in others (Fig. 1b). Next, we will analyze and discuss the first set of 8 cases listed in Table 1 to understand the role of different medium SE on coupled model PE.

## 3.1 Stability, reliability and convergent rate of PE with partial SE

In Table 1, "X-to-Y" means using observations of "X" to estimate the parameter "Y" ("$x_2$-to-$a_2$" means using observations of $x_2$ to estimate parameter $a_2$, for instance). Table 1 shows that all 4 PE cases with atmospheric SE succeed while all 4 PE cases with oceanic SE fails, no matter what medium observations are used to estimate which medium parameter. The root mean square error (RMSE) of the model states and parameters during the last 100 TUs are shown in Table 2. The $x_2$ RMSE in the failed cases are higher than in the successful cases, while the $w$ RMSE in two failed cases F(5) and F(6) are even smaller than the ones in successful cases. Also, in both cases of S(2) and F(6), $a_2$ is estimated by $w$ observations but only when the $x$ states are constrained by $x_2$ observations, the PE is successful, or otherwise the PE is failed although the state $w$ is constrained by the $w$ observations. These suggest that the uncertainty of $x_2$ is mainly responsible for the failure of the PE. An example of failed PE in which the observations of $w$ are used to estimate $a_2$ is shown in Fig. 1b. We can see the ensemble of $a_2$ in Fig. 1b cannot converge to its "true" value of 28. We will thoroughly analyze such failed cases next.

The stability of PE is different among partial SE settings as shown in Figs. 2 and 3 as the time series of the ensemble mean of the estimated parameters. Figs. 2bc and 3bc show the 4 successful cases with only atmospheric SE. Compared to full SE (using observations of $x_2$ and $w$, shown in Figs. 2a and 3a), the partial SE cases show much bigger fluctuation in estimated parameter values at the beginning of spin-up period (Figs. 2bc and 3bc). From Figs. 2 and 3, it can also be seen that generally the accuracy of PE with partial SE is lower although overall the estimated parameter values converge to the truth. This can be comprehended by the lower signal-to-noise ratio of state-parameter covariance provided by the SE process, which will be discussed more details at the end of this section.

The convergence rate of PE is also obviously different with different SE settings. The case of $w$-to-$a_2$ converges much more slowly than the other cases in $a_2$ estimation. This phenomenon can be explained by the different time scales of different media. Figure 4 shows the variation of the state variable during SE. The observational constraint makes the mean value and the whole ensemble to follow the "truth" (see Fig. 4a for $x_2$ and Fig. 4e for $w$). It can be seen that in cases assimilating $x_2$, due to no direct constraint on $w$ and $\eta$, their spread shrinks slowly. Instead they are forced by the constrained $x_2$ but with slower adjustment of ocean processes. As mentioned in section 2.3, the SE starts before the PE to make sure the state needed is constrained enough. Slowly shrinking of $w$ and $\eta$ spreads shall be considered in determining a longer delayed time for the PE related to $w$ and $\eta$.

The inflation method is also important in PE (Yang & DelSole, 2009; DelSole & Yang, 2010; Zhang, 2011ab; Zhang et al., 2012). The partial and full SE cases are with the same inflation scheme (Zhang, 2011ab; Zhang et al., 2012). Shadows in Figs. 1-3 show the range of the parameter ensemble. The zigzag shape of the shadows represents the inflation during PE. In these figures, the width of the shadows shrinks quickly once PE is activated while some of the mean values moves toward

the "truth" slowly (for example, Fig. 2c and Fig. 3b). Also from the zigzag shapes, we can see some inflation effects before the parameter converges to the "truth." All these imply that the designed PE is stable and its convergence rate is not much sensitive to the inflation scheme.

In addition, larger ensemble sizes are used to test the sensitivity of the conclusion above. The results show that bigger ensemble size has a positive impact on SE and PE quality but the drawn conclusion from the experiments above does not change its essence. Also the ensemble size far exceeds the problem size in this simple model study. In this regard, further examination may be necessary in CGCM cases. We also performed the experiments under different SE update interval settings. Test results show that for the issue we are addressing, the conclusion is not sensitive to the update interval if it is within a reasonable range ($x_2$-to-$c_2$ and $w$-to-$c_2$ fail on any update interval with SE of $w$ and succeed with SE of $x_2$ in a SE interval range of no larger than 0.3 TUs, for instance).

In cases 3 and 4, we successfully estimate the oceanic parameter $c_2$, suggesting we can use different medium measurements to help calibrate the parameter within a coupled model. In case 3, the atmospheric observations are used for both SE and PE, while in case 4, the atmospheric observations are used for SE but the oceanic observations are used for PE. The case 3 uses only the atmospheric observations to determine an oceanic parameter and does a better job than using the oceanic observations does in case 4.

The phenomenon above in estimation of $c_2$ can be comprehended by the "air-sea" interaction process. What about a pure oceanic parameter (a parameter used for deep ocean, for instance)? It is interesting to see the influence of atmospheric SE accuracy on PE for a deep ocean parameter. To do that, a series of $\eta$-to-$c_6$ PE experiments with different SE settings is carried out. The deep ocean observation is generally sparse in the real world. But within our twin experiment framework described in section 2.3, the "observations" of $\eta$ used for our PE can be produced as sufficient as other variables. All PE experiments on $c_6$ are conducted with $\eta$ observations (observations of $x_2$ and $w$ are only used in different SEs but not used in the PE). The result is shown in Fig. 5. Given the long time scale of $\eta$, the $\eta$ PE experiments are extended to $10^4$ TUs. The PE cases include 4 SE settings. They are case-1: all state variables, case-2: $x_{1,2,3}$ only, case-3: $w$ and $\eta$, and case-4: $\eta$ only. Both case-1 and case-2 succeed greatly, but the convergence rate of case-1 is faster than case-2 and the accuracy of case-1 is a little higher than case-2. In case-3, the convergence rate is fast but the estimated values remain in a bias from the truth. Case-4 apparently fails, never stably converging to any value. It is clear that the $\eta$-to-$c_6$ PE succeed only when the atmospheric state is constrained by observations.

It is interesting that once the atmospheric states (the Lorenz equation in this simple model) are constrained by the observations, both the atmospheric parameter ($a_2$) and oceanic parameters ($c_2$ and $c_6$) can be successfully estimated even in the case using the atmospheric observations ($x_2$) to estimate the oceanic parameter ($c_2$) or using the ocean observations ($w$) to estimate the atmospheric parameter ($a_2$). This seems different from our previous intuition that in-situ ocean data are always considered as the first important piece of information for determining the oceanic coefficients. Our results here strongly

suggest that in the future real coupled model PE experiments, for determining the best coefficient values, no matter the atmospheric or oceanic, sufficient and accurate atmospheric measurements are crucially important. Next we will conduct more sophisticate analyses to extend our understanding on this point.

In our twin experiment setting, there are 3 types of model uncertainties: strong nonlinearity in the atmosphere (chaotic in this case), weaker nonlinearity in the ocean and biased parameter values. The SE process before PE aims to control the first and second types of the uncertainties by observational constraints on model states. Figure 6 shows the wavelet analyses for the atmospheric variable $x_2$ and the oceanic variable $w$ in the "truth" run. They represent the uncertainties of type 1 (panel a) and type 2 (panel b). With the expanded exhibition of wavelet on different periods, Fig. 6 clearly tells significantly different features of $x_2$ and $w$. The energy of $x_2$ is in the high frequency band and the energy of $w$ is in the low frequency band. $x_2$ varies fast and represents the most uncertain mode, transferrable to low frequency $w$ through the "air-sea" interaction. Later in section 3.2, we will show that the feedback of ocean can magnify the role of atmospheric chaotic forcings. The chaotic nature can spread out and results uncertainties in all frequency band in the system. Under such a circumstance, the method of picking a particular frequency (e.g. Barth et al., 2015) or using averaged covariance (Lu et al., 2015) to implement PE cannot essentially resolve the issue although it may relax the problem. Instead, reducing $x_2$ uncertainty (enhancing the estimation accuracy of the atmospheric states) is more relevant to the solution of the problem.

Without direct observations on parameter values, PE completely relies on the covariance between the parameter and model states for projecting the observational information of states onto the parameter. While the PE projection is carried out by a linear regression equation based on the state-parameter covariance (EnKF/EAKF, for instance), only a linear or quasi-linear relationship between parameters and states in ensemble is recognized. All failure of PE without direct atmospheric SE could be attributed to the chaotic disturbances in the atmosphere (Lorenz equations in this case) that make difficulties for the system to build up a quasi-linear relationship between the state variable and the parameter.

To investigate the parameter-state relationship in the model background (prior PE), we conduct a series of parameter perturbation runs corresponding to 8 partial SE experiments (without PE to fix the parameter spread – the PE process sets the parameter ensemble as an additional system freedom and makes the relationship of the parameter and model state more complicate). In that way, the parameter perturbations can be fully transferred to the model states so that we can study the state-parameter relationship in a straight forward manner. The results are shown in Figs. 7 and 8, where the horizontal axis is the ensemble anomaly (vs. ensemble mean) of the state variable and the vertical axis is the ensemble anomaly of the parameter, and the background black dots represent the model runs starting from different initial conditions. Since the parameter ensemble does not change (once perturbed at the initial time) during the model integration, the lines constructed by black dots in a perturbation run are parallel to the x-axis perfectly. However, the set of dots at the same integration time step from different initial conditions can be used to sample the relationship between the perturbed parameter and the model state. For example, 2 sets of such ensembles, which have the biggest positive and negative correlation coefficients between the parameters and the model states, are coloured (20 red dots and 20 blue dots) in each case. From Fig. 7, we can see that

with SE for the atmosphere, the overall quasi-linear relationship between the model state anomalies (observational increments) and the parameter adjustments is constructed by the model. Under this circumstance, a meaningful projection from the observational increment on the parameter is gained to form a signal-dominant adjustment for the parameter ensemble. As shown in Fig. 8, without the atmosphere SE, the linear relationship between the parameter being estimated and the model states is not correctly built up, and thus the parameter estimation fails.

Relationship between the states and the parameters can be analyzed quantitatively. Zhang et al. (2012) defined an ad hoc index to measure the signal-to-noise ratio (called $r_{s2n}$) of a model ensemble. Following the idea, we diagnose the signal-to-noise ratio of the ensemble-based error covariance between the states and parameters here. The new $r_{s2n}$ is defined as $R \times S$, where R is the averaged correlation coefficient between the parameter perturbations and the ensemble states in a selected time window, and S is the ratio of root mean square linear fitting errors of the parameter-state points in the full SE and in a partial SE ($S_f/S_p$). The best (worst) representation of the signal-to-noise ratio is then characterized by a $r_{s2n}$ value of 1(0). Table 3 gives the $r_{s2n}$ values for the SE only experiments of Fig.7 and Fig.8. Correlation coefficients of F(5) and F(8) are 0.19 and 0.24 respectively. Though the dependences of $x_2$ on $a_2$ in F(5) and $w$ on $c_2$ in F(8) are fairly direct, the low R values suggest these relations can be easily interrupted by the atmospheric uncertainty. The values of $r_{s2n}$ are much higher in the successful cases than in the failed cases. These results clearly show that reduction of the atmospheric uncertainty can greatly increase the signal-to-noise ratio of the parameter-state covariance in the system through enhancing the bonding between the state variable and the estimated parameter.

### 3.2 Impact of the chaotic-to-periodic ratio in forcings on oceanic PE

From the results above, we learned that the PE of $c_2$ or $c_6$ strongly relies on the SE of $x$. In a coupled system characterized as Eq. (1), the influence of atmosphere can thoroughly propagate to all variables of other media, although the influence may reduce for the deep ocean. However, some previous studies (e.g. Annan et al., 2005; Barth et al., 2015; Gharamti et al., 2014; Leeuwenburgh, 2008; Massonnet et al., 2014) show their successfulness in estimating parameters in ocean only using oceanic observations without constraints on atmospheric states. To understand what character of the model makes this difference, we make full use of this simple model with convenience to investigate the influence of model characteristics on coupled parameter estimation. For mimicking the real climate signals, the variability of the oceanic state variables $w$ and $\eta$ in Eq. (1) are driven by two kinds of forcings: the chaotic forcing from the atmosphere (Lorenz equations) and the periodic forcing associated with the external radiative forcing (simulated by a cosine function with the amplitude coefficient of $S_s$ in this simple model). The oceanic states in the real world consist of both periodic and chaotic variations. The periodic characteristic of a state is naturally with high predictability and is generally easier to be detected after an averaging or filtering process. In this simple model, $w$ ($\eta$) is directly under the influence of the parameter $c_2$ ($c_6$) - perturbations of $c_2$ ($c_6$) first directly affecting $w$ ($\eta$) and then influencing the whole model by the interactions between $w$ ($\eta$) and other variables. To understand the influence of chaotic/periodic variability of the ocean on oceanic parameter estimation, we modify the model in Appendix A to set one-way coupling model, i.e. only $w$ is forced by $x_2$ but $x_2$ remains independent from $w$. In that way we

do not need to worry about the instability of Lorenz equations due to the dramatic influence from large $w$ values. Then we define a chaotic-to-periodic ratio (CPR) in the signals of $w$ ($\eta$) to study the PE performance under different chaotic/periodic variability regime of a model system. Details of the CPR definition is given in Appendix B. The CPR of $w$ ($\eta$) can be easily manipulated by changing the coefficient of $S_s$. We first compare the results of $w$ in one-way coupling (Fig. 9a) and two-way coupling (Fig. 6b) models with the identical $S_s$ value of 1. The CPR in the full period (from 0.3 TUs to 165 TUs, the longest period that is selected to avoid boundary effects) of $w$ in Fig.9a is 1.0963. It's interesting that the one-way coupling (without $w$'s feedback to $x_2$ in the model) can transfer more energy to low-frequency band. Then we perform eight PE experiments, four for $w$-to-$c_2$ and four for $\eta$-to-$c_6$. We examine 4 $S_s$ values of 100, 250, 500 and 1000, representing a reduced CPR sequence of $w$ ($\eta$). Their CPR values are respectively 1.0485 (0.6084), 1.0386 (0.6083), 1.0333 (0.6081), 1.0282 (0.6080). Note that the CPR value will change once a PE process is activated. We compare these one-way coupling model results and show two examples ($S_s$=1 and $S_s$=250, Figs. 9ab for w, Figs. 10ab for $\eta$). We found that the increasing amplitude of periodic forcing can enhance the periodic signals for $w$ and $\eta$. Clearly, when the $\eta$ CPR decreases, the periodic portion dominates and the $\eta$-to-$c_6$ PE becomes more and more robust (see Figs. 11a-d). But in the other 4 $w$-to-$c_2$ cases, for any $w$ CPR, the $w$-to-$c_2$ PE fails (Fig. 12a). This is due to strong dependence of cov($w$, $c_2$) (the covariance between $w$ and $c_2$) on $x_2$ (see Eq. (1)) that is still chaotic without observational constraint. Though $w$ is very periodic, the chaotic variability of $x_2$ sheds on $w$'s variability (the needed variability of $w$ for PE should come from $c_2$ but now comes from the chaotic $x_2$) and makes the PE process misjudge the difference between the simulated $w$ and its observation, thus not producing a correct PE projection.

To further test the role of periodic signals in ocean states for oceanic PE, we conduct oceanic PE on a particular frequency band using the method described in Appendix C. Some results are shown in Fig. 12 which tells that using the covariance of $\eta$ in a particular frequency and $c_6$ to project the corresponding $\eta$ observational information can make a $\eta$-to-$c_6$ PE case with $S_s$ = 250 as successful as the result of $S_s$=1000 with full frequencies (compare Fig. 12b to Fig. 11d). The method is designed to limit the PE process working on the 10 TUs period of $\eta$ information, which dramatically reduces the CPR of $w\eta$ (the CPR of $\eta$ now is 0.1424, and the CPR of $w$ is 1.0525 at the beginning of the PE) and thus helps $c_6$ estimation, but given strong dependence of cov($w$, $c_2$) on $x_2$, and that the CPR of $x_2$ is big on every frequency band, this particular frequency PE method does not help for estimation of $c_2$ (Fig. 12a).

**4 Conclusion and discussions**

The erroneous values of parameters in a coupled model are a source of model bias that can cause model climate drift. Model bias can be mitigated by parameter estimation (PE) with observational data. The signal-to-noise ratio in state-parameter covariance plays a centrally important role in the PE process. With a conceptual coupled model, we discuss the issue how to enhance the signal-to-noise ratio in coupled model PE through further understanding on various aspects of the PE process in a coupled numerical system.

We performed 3 kinds of comparisons to discuss the issue. The first kind focuses on the PE performance with a two-way coupling model. Results show that atmospheric state estimation (SE) is critically important. The second comparison is carried out by the experiments with the same parameter spread and SE settings in the first comparison but without the PE process. We use this way to examine the signal-to-noise ratio of state-parameter covariance in different SE settings. Results find that the projection of the observational increment onto the parameter can be easily interrupted under partial SE conditions. In the third kind, we changed the model structure from two-way coupling to one-way coupling, allowing the ocean state varying forced by the atmosphere without feedback to the chaotic atmosphere. The PE results are better with higher periodic and less chaotic states.

According to all these comparisons, first, we found that due to the interaction of multiple time scales in our conceptual coupled model, the fast varying component is the major source for producing an inaccurate state-parameter covariance in the system. Enhancing the estimation accuracy of high-frequency states that interact with the parameter is the most important to maintain a signal-dominated relationship between the parameter being estimated and model states, and makes successful coupled model parameter estimation. Second, the chaotic-to-periodic ratio (CPR) of the model state that closely associates with the parameter being estimated determines the required state estimation accuracy. Given limited observational resources, in the future when we work with realistic model and observing system, the CPR shall be first investigated to increase the opportunities of having successful parameter estimation.

Given the fact that observations are always imperfect, this conceptual coupled model study tries to provide some general guideline for CGCM PE application with the real observing system. However, the results have the following limitation: 1) The conceptual coupled model assumes that only the atmosphere is a chaotic uncertainty source. In the real world this is unnecessarily true (nonlinearity produced by smaller scale eddies in the ocean could be the part of chaotic uncertainty sources too, for instance). 2) The atmosphere-ocean interaction is idealized in the conceptual model. In the real world, the air-sea coupling could be complex as highly geographic dependent. 3) The twin experiment assumes that except for the parameters to be estimated, the model "dynamical core" and "physics" are perfect and consistent to the "observation." In the real world, the CGCM is biased from the observations. All these aspects still need to be addressed before coupled model PE is applied to a CGCM with the real observing system.

How the accuracy of state estimation impacts on the coupled model parameter estimation is an interesting and challenging research topic. The spatial and temporal dependence of atmospheric and oceanic circulations could further complicate the issue. For example, the Kuroshio meander in the south of Japan is very different to the Kuroshio meander cross the Luzon strait. The Kuroshio cross Luzon strait is easily interrupted by the monsoon, but the meander in the south of Japan is a self-sustained dynamic system having multiple equilibria with non-periodic state changes (Taft, 1972; Yu et al., 2013); the uncertainty of the latter comes from the accumulation of the negative vorticities in the ocean. Further, we have already known that the method on a particular frequency can increase the opportunity of successfulness. When such a real problem is addressed through the PE with a CGCM, we may need to make efforts on both adaptive measurements and spectral separation. The PE method shall be improved to perform separately at different time scales. How to speed up the

convergent rate in the coupled model PE process is also an important issue. All of these require further research work to clarify.

## Appendices

### Appendix A: One-way coupling model

A suitable scope of parameter values that maintain the model character is an important pre-condition for successful PE. For example, in Eq. (1) when $a_2$ is lower than 20, the variation of $x_2$ becomes periodic and looses the chaotic nature. When the values of the parameter of some ensemble members are numerically out of bound, different ensemble members exhibit different dynamic performance (some of them are chaotic and the rest are periodic), and the state-parameter covariance computed from the ensemble becomes unreasonable and PE must fail. In $a_2$ PE experiments, the values are bounded within

24 ~ 32 where nonlinearity and characteristic variability of the model maintains. For the purpose of manipulating the signal $w$ or $\eta$, to make them become more periodic than chaotic, we changed the parameter $S_s$ to magnify the amplitude of the cosine term that directly forces $w$. This causes the value of $w$ to grow bigger according to different $S_s$ settings. At the same time, the original two-way coupling has to be changed to one-way coupling by removing the $w$ in the $x_2$ equation, which interacts with $a_2$ in the Lorenz equation, for keeping the ability of producing the chaotic signal. The referring $x_2$ equation

after the modification is:

$$\dot{x}_2 = -x_1 x_3 + (1 + c_1) a_2 x_1 - x_2 \tag{A1}$$

Therefor, when using Eq. (A1), the Lorenz atmosphere cannot feel the variation of the ocean. The strength of the chaotic forcing remains the same in all cases with different $S_s$ settings. And because the Lorenz atmosphere runs independently, there are no needs to set scope limits of the oceanic parameter $S_s$, $c_2$ and $c_6$ for securing the chaotic character of the system

under this circumstance. The oceanic parameters can be perturbed much larger than in the two-way coupled cases.

### Appendix B: Definition of CPR

Chaotic naturally lowers predictability of the signal. The chaotic-to-periodic ratio (CPR) is defined to measure the chaotic degree of a system within a particular period band as:

$$CPR = \frac{1}{T_2 - T_1} \int_{T_1}^{T_2} \text{std}\{\log_2[P(T) + 1]\}\, dT \tag{B1}$$

Here $P$ is the wavelet power spectrum of the selected state variable on the period of T, and std denotes the standard deviation (of the base-2 logarithm of $P$ performed along a time window, plus 1 to ensure positive definite for the logarithm result). The wavelet transformation is able to identify period components simultaneously with their location and time. The CPR is a positive definite indicator. Its value is 0 for a pure periodic signal.

**Appendix C: PE method on a particular frequency band**

Previous studies have shown that applying the PE with an averaged covariance in particular time window can increase the signal-to-noise ratio (Lu et al., 2015, Barth et al., 2015). In our case, it can also effectively increase the CPR of the state variable. Here, we propose an alternative method that has similar effect as an averaged covariance but is much easier to be implemented. This method applies PE on a particular frequency. The method succeeds to enhance the CPR by using a designed filter on both the observations and the simulated ensemble results, and it can allow information focusing on a particular frequency more accurately than using the averaging method.

In this study, for the $\eta$-to-$c_6$ PE case with $S_s$=250, the periodic signal produced by the cosine function has a period of 10 TUs (1000 time steps) (defined by $S_{pd}$ in Eq. (1), also see Fig. 10) and the chaotic signal is much slower than the periodic signal. In other words, the signal/noise ratio of $\eta$ is strongest on the period of 10 TUs. Therefore we designed a Butterworth high pass filter (BF) with a frequency pass band equal and larger than Fs/1000 (Fs is the frequency of sampling) to help the PE of $\eta$-to-$c_6$. The parameter update interval in the new PE method is identical to the standard full frequency PE case, but for each update step, before applied to Eq. (2) and Eq. (3), the observation and simulated ensemble results are filtered by the following BF process:

$$
\begin{aligned}
&\text{old}: \Delta y_i^o = \text{PE}(y^o, y_i^p) \\
&\text{new}: \Delta y_i^o = \text{PE}\left[ \text{Filter}(y^o), \text{Filter}(y_i^p) \right]
\end{aligned} \quad , \quad (\text{i=1~N, N is the ensemble size}) \tag{C1}
$$

Here $y^o$ is the observation and $y_i^p$ represents the simulated ensemble results. The BF is applied within a 5000 steps (or more) moving window. It means that on each PE step, the last 5000 observations and the simulated ensemble results in the same window are transformed through the same BF to produce new observations (Hobs) and new simulated results (Hens) on the particularly frequency. Then the new $\Delta y_i^o$ is computed from the Hobs and Hens, and it is used with the covariance to determine the adjustment of the parameter. This new method can be used for different frequency band (low-pass, high-pass or band-pass), it succeed to improve the PE performance in our one-way coupling experiment for the $\eta$-to-$c_6$ PE (Fig. 12b).

**Acknowledgements**

This work is funded by the National Natural Science Foundation of China (41306004), the China's National Basic Research Priorities Programmer (2013CB956202) and the National Natural Science Foundation of China (41490640;41490641).

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

**Table 1: List of the successful (S) and failed (F) parameter estimation (PE) cases with partial state estimation (SE) in 8 PE experiments (in the parenthesis is the experiment serial number).**

| SE \ PE | $x_2$-to-$a_2$ | $w$-to-$a_2$ | $x_2$-to-$c_2$ | $w$-to-$c_2$ |
|---|---|---|---|---|
| $x_{1,2,3}$ by $x_2$ obs | S (1) | S (2) | S (3) | S (4) |
| $w$ by $w$ obs | F (5) | F (6) | F (7) | F (8) |

**Table 2: List of root mean square error of the state variable and the parameter during the last 100 TUs in 8 PE experiments.**

| Exp number \ State and Param | $x_2$ | $w$ | $a_2$ | $c_2$ |
|---|---|---|---|---|
| S(1): $x_2$ obs, $x_2$-to-$a_2$ | 5.9224 | 0.0570 | 0.0889 | N/A |
| S(2): $x_2$ obs, $w$-to-$a_2$ | 5.9086 | 0.0567 | 0.0895 | N/A |
| S(3): $x_2$ obs, $x_2$-to-$c_2$ | 5.9213 | 0.0731 | N/A | 0.0250 |
| S(4): $x_2$ obs, $w$-to-$c_2$ | 5.9174 | 0.0589 | N/A | 0.0153 |
| F(5): $w$ obs, $x_2$-to-$a_2$ | 14.6801 | 0.0360 | 1.6806 | N/A |
| F(6): $w$ obs, $w$-to-$a_2$ | 14.3177 | 0.0381 | 3.2612 | N/A |
| F(7): $w$ obs, $x_2$-to-$c_2$ | 14.4102 | 0.0744 | N/A | 0.3848 |
| F(8): $w$ obs, $w$-to-$c_2$ | 14.4004 | 0.0660 | N/A | 0.3454 |

**Table 3: List of $r_{s2n}$ during the last 100 TUs in 8 SE only (no PE) experiments.**

| | S(1): $x_2$ obs, $x_2$-to-$a_2$ | S(2): $x_2$ obs, $w$-to-$a_2$ | S(3): $x_2$ obs, $x_2$-to-$c_2$ | S(4): $x_2$ obs, $w$-to-$c_2$ | F(5): $w$ obs, $x_2$-to-$a_2$ | F(6): $w$ obs, $w$-to-$a_2$ | F(7): $w$ obs, $x_2$-to-$c_2$ | F(8): $w$ obs, $w$-to-$c_2$ |
|---|---|---|---|---|---|---|---|---|
| R | 0.41 | 0.33 | 0.60 | 0.91 | 0.19 | 0.19 | 0.19 | 0.24 |
| $S_f/S_p$ | 0.65 | 0.97 | 0.98 | 0.96 | 0.19 | 0.96 | 0.29 | 0.83 |
| $r_{s2n}$ | 0.27 | 0.32 | 0.59 | 0.87 | 0.04 | 0.18 | 0.06 | 0.20 |

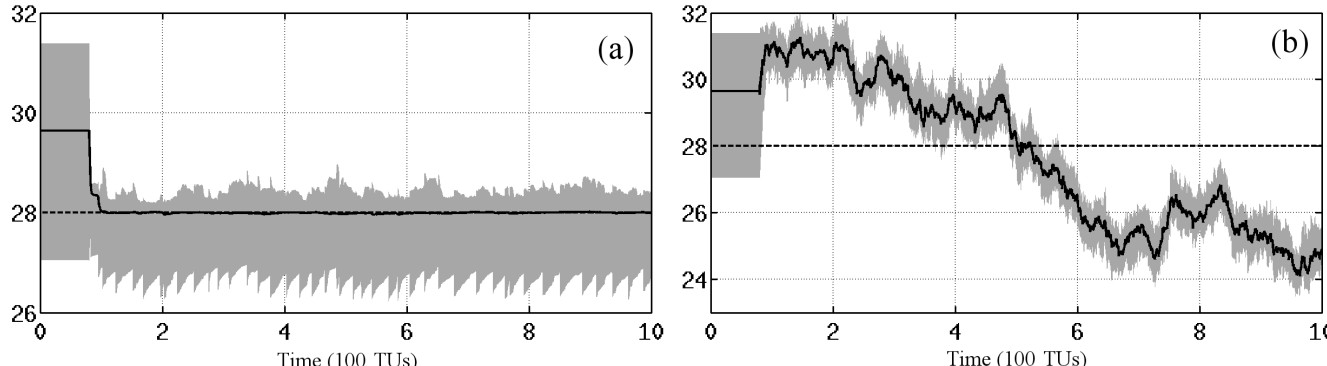

**Figure 1:** Time series of the ensemble mean (solid line) of the estimated parameter $a_2$ using observations of $w$ (i.e. $w$-to-$a_2$) with state estimation (SE) of a) both the atmosphere ($x_{1,2,3}$) and ocean ($w$) from $x_2$ and $w$ observations, and b) only $w$ with the $w$ observations. The dashed line marks the "true" value of the parameter $a_2$ and the shaded area represents the range of ensemble.

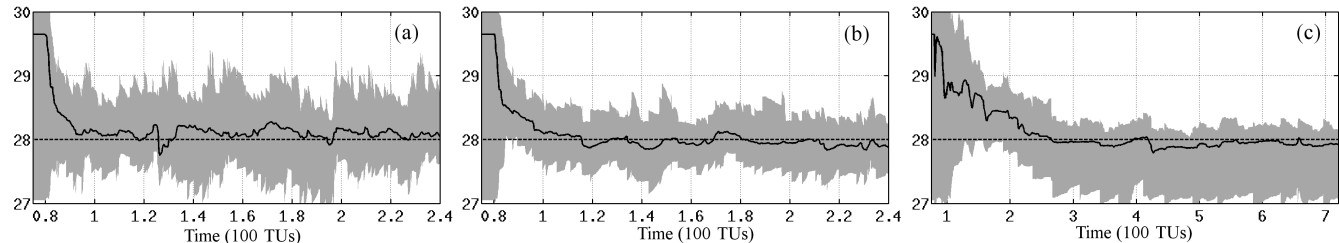

**Figure 2:** Time series of ensemble means (solid line) of the estimated parameter $a_2$ in 3 experiments, a) $x_2$-to-$a_2$ (using $x_2$ observations to estimate $a_2$) with SE for both $x_{1,2,3}$ and $w$, b) $x_2$-to-$a_2$ with SE for $x_{1,2,3}$ only, c) $w$-to-$a_2$ with SE for $x_{1,2,3}$ only. Any other notations are the same as in Fig. 1.

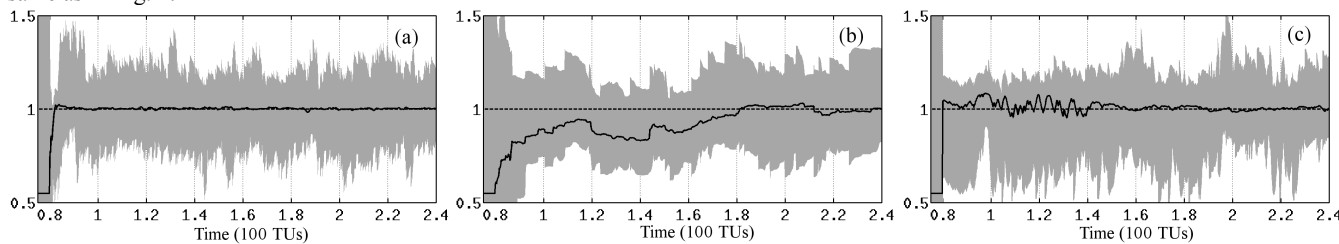

10 **Figure 3:** Time series of ensemble means of the estimated parameter $c_2$ in 3 experiments, a) $w$-to-$c_2$ (using $w$ observations to estimate $c_2$) with SE for both $x_{1,2,3}$ and $w$, b) $x_2$-to-$c_2$ (using $x_2$ observations to estimate $c_2$) with SE for $x_{1,2,3}$ only, c) $w$-to-$c_2$ with SE for $x_{1,2,3}$ only. Any other notations are the same as in Fig. 1.

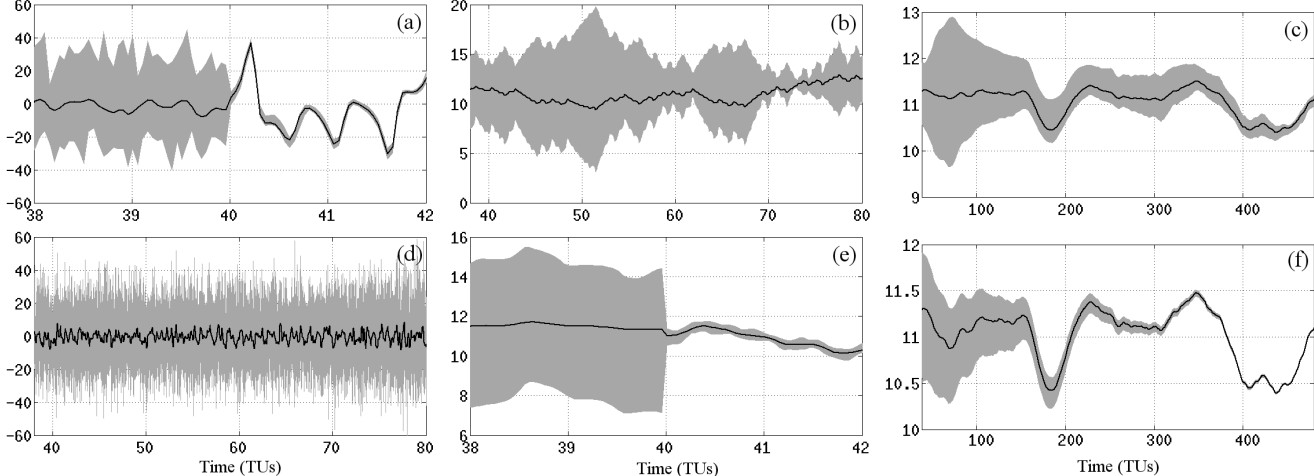

**Figure 4:** Time series of the state variables from the $w$-to-$c_2$ PE experiment, for ad) $x_2$, be) $w$ cf) $\eta$. The upper panels abc) are from the successful case with SE for $x_{1,2,3}$, and the lower panels def) are from the failed case with SE for $w$. Any other notations are the same as in Fig. 1.

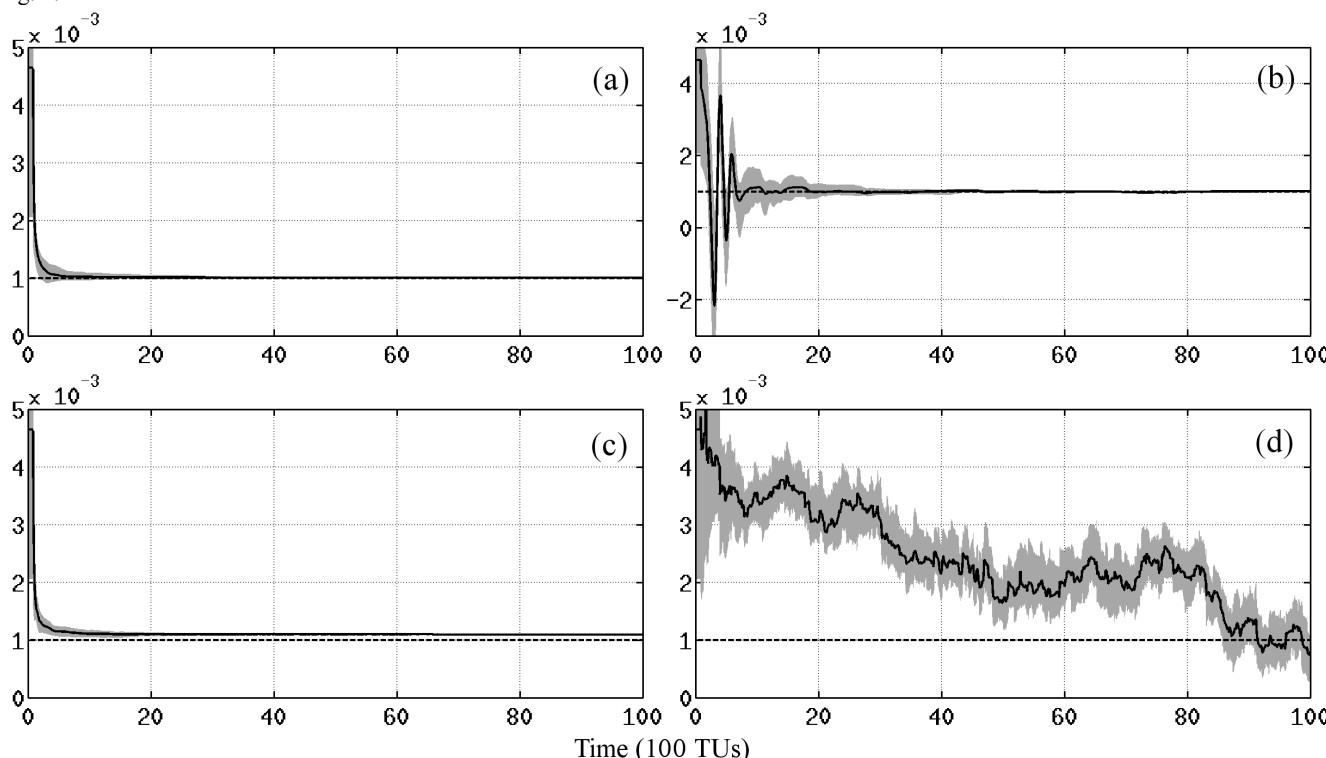

**Figure 5:** Time series of the ensemble of parameter $c_6$ from the $\eta$-to-$c_6$ (using $\eta$ observations to estimate $c_6$) PE experiment in 4 different state estimation settings, a) $x_{1,2,3}$, $w$ and $\eta$, b) $x_2$ only, c) $w$ and $\eta$ only and d) $\eta$ only. Any other notations are the same as in Fig. 1.

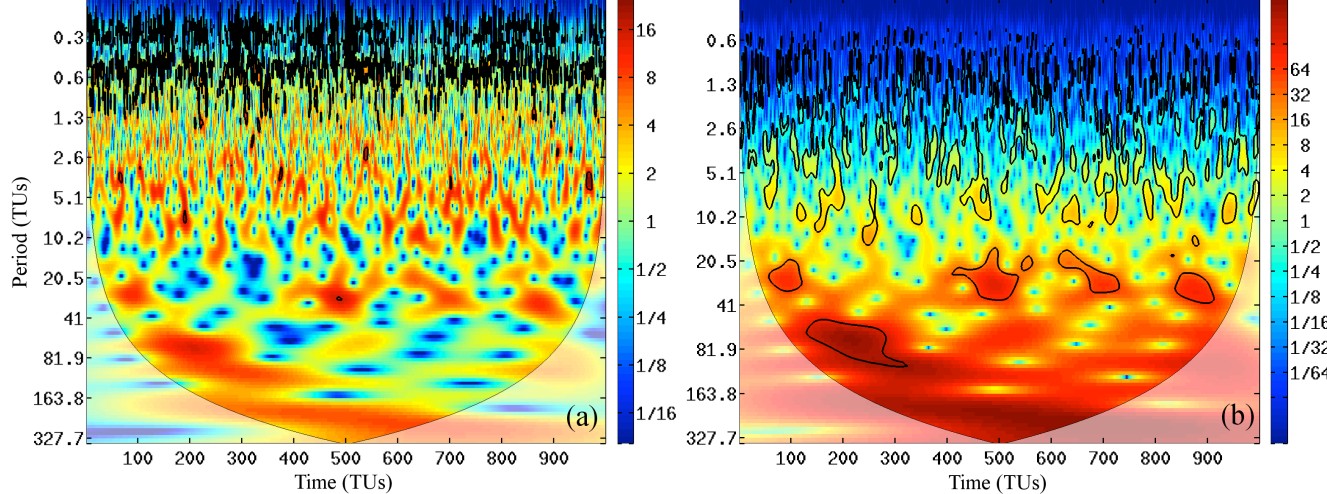

**Figure 6:** Wavelet analyses for a) $x_2$ and b) $w$ in the "truth" model run.

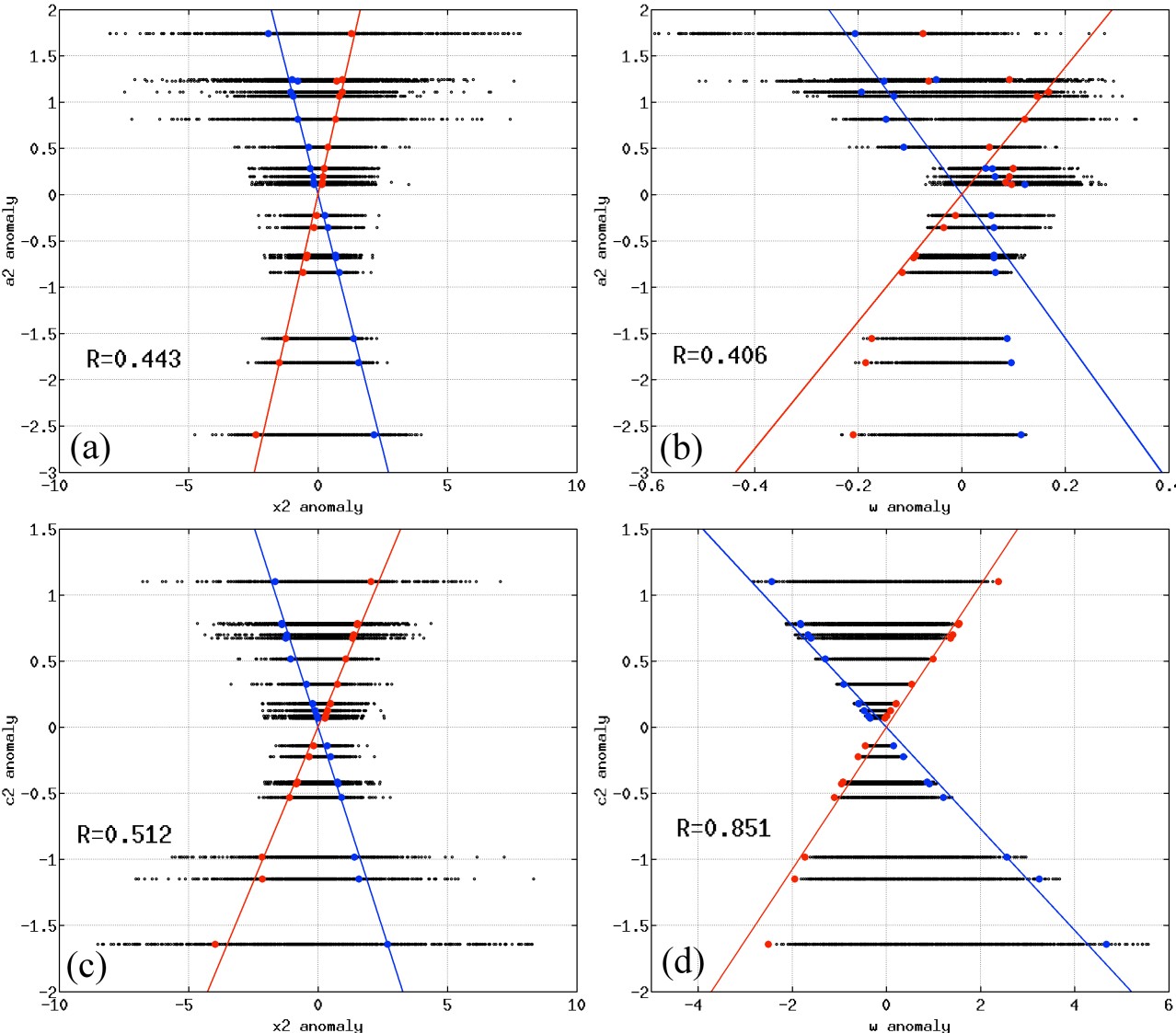

**Figure 7:** Sampling map of the perturbed parameter anomalies in the space of model state anomalies for a) $a_2$ vs. $x_2$, b) $a_2$ vs. $w$, c) $c_2$ vs. $x_2$ and d) $c_2$ vs. $w$, when the atmospheric state is constrained by its observations. Dots with the same color (red or blue) represent ensembles at the same time step in the model integration. The colored line represents a linear fitting for the same color dots. Here we show two examples that have a high positive (red) and negative (blue) correlation between the parameter and model state perturbations, respectively. The R value shown in each panel is the time averaged parameter-state correlation coefficient in last 5000 time steps.

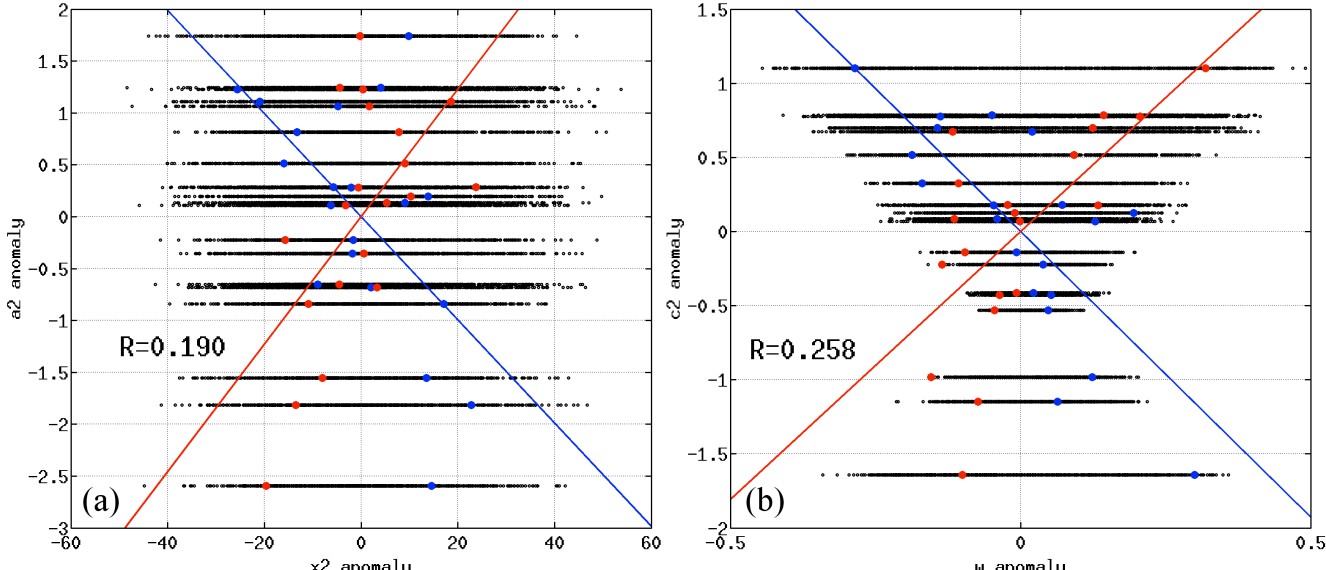

**Figure 8:** Same as Fig. 7 but for the case with SE of $w$ only, a) $a_2$ vs. $x_2$ and b) $c_2$ vs. $w$. Here we show two examples that the linear fitting becomes difficult in red and blue, for which the data are taken from the same time steps as shown in Fig. 7.

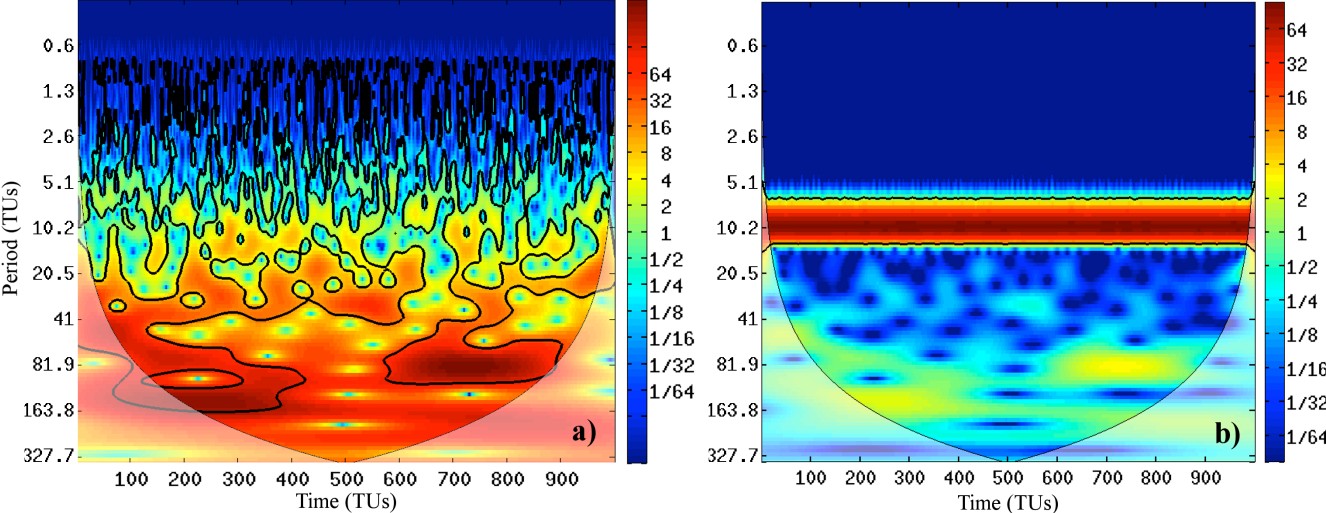

5    **Figure 9:** Wavelet analyses for $w$ in the run of one-way coupling model forced by a) $S_s$=1 and b) $S_s$=250.

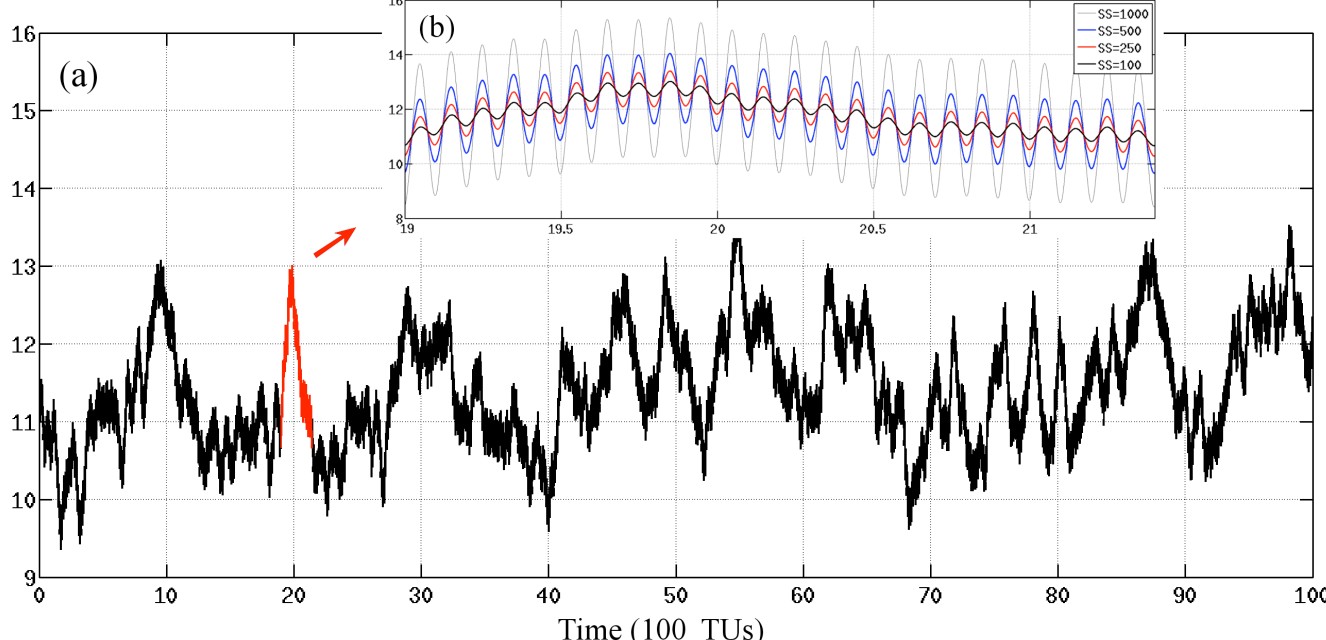

**Figure 10:** Time series of $\eta$ with different $S_s$ values (varying from 100 to 1000) with a one-way coupling model setting described in Appendix A. To visualize the difference induced by different $S_s$ values, panel b) is the zoomed out version of the section marked in red in panel a).

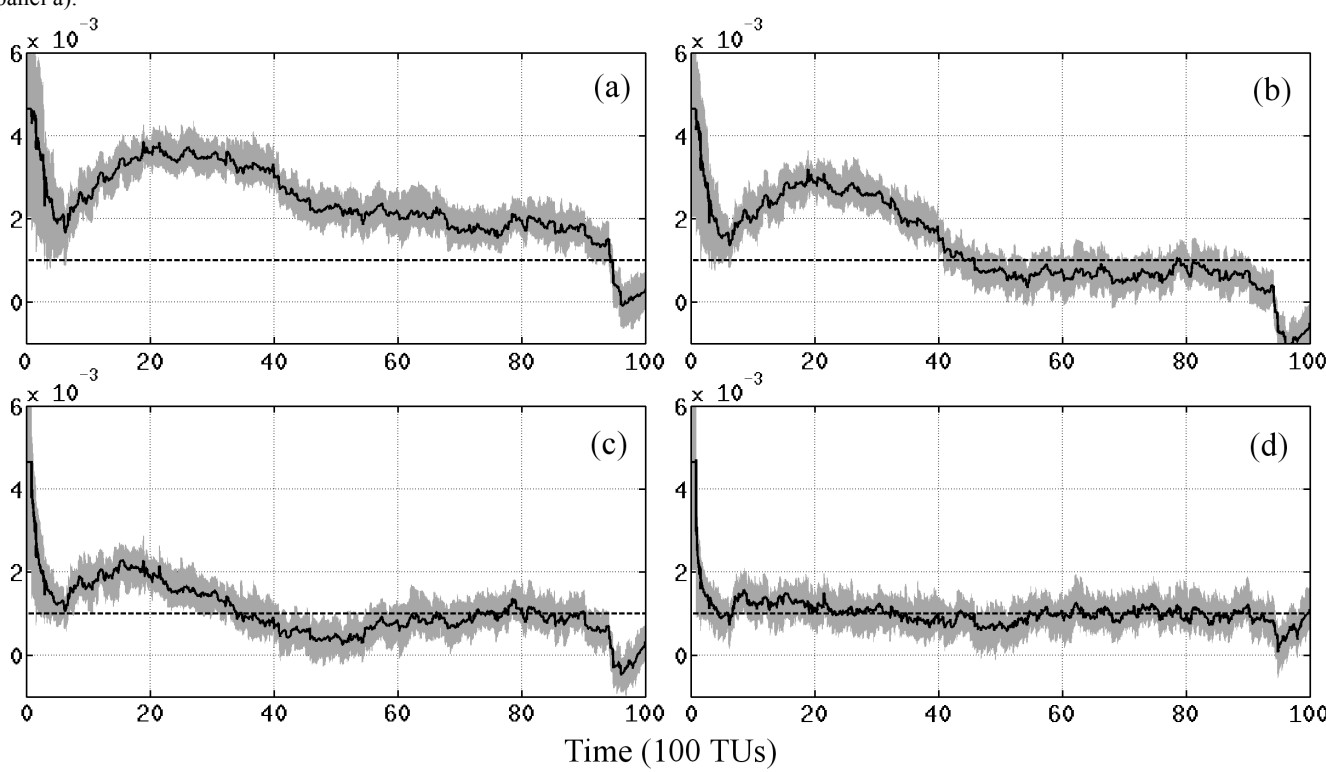

**Figure 11:** Time series of the ensemble of parameter $c_6$ in 4 $\eta$-*to*-$c_6$ PE experiments with different $S_s$ values, a) 100, b) 250, c) 500 and d) 1000 with the one-way coupling model setting. In all cases, only $\eta$ is constrained by its observations. Any other notations are same as Fig. 1.

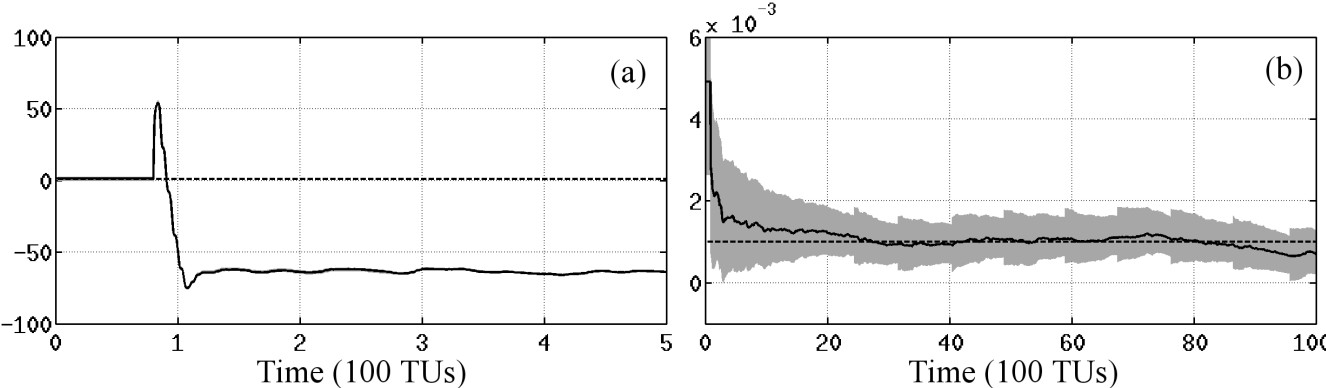

5    **Figure 12:** Time series of the ensemble of the parameter in the a) $w$-to-$c_2$ PE with SE of $w$ only and b) $\eta$-to-$c_6$ PE with SE of $\eta$ only using the one-way coupling model with $S_s$=250. Note that the initial $c_2$ in panel a) is approximate 0.56, and the truth is 1. Any other notations are same as Fig. 1.

