# Peer review of "Insights on the Role of Accurate State Estimation in Coupled Model Parameter Estimation by a Conceptual Climate Model Study"

_Nonlinear Processes in Geophysics, 2016_

## Referee Comment (RC1) · Anonymous Referee #1 · 8 Nov 2016

**Review of the manuscript "Further Insights on the Role of Accurate State Estimation in Coupled Model Parameter Estimation by a Simple Climate Model Study" by X. Yu et al.**

The manuscript investigates the impact of observational constraints, through data assimilation methods, on coupled model state and parameter estimation using a conceptual 5-variable model.
I found the manuscript interesting and appropriate for the journal, especially to fill the existing gap of idealized studies in coupled data assimilation experiments. It is somehow less relevant, in my opinion, for parameter estimation, given the complexity of real-world CGCM, as the authors themselves discuss in the Conclusions.
I recommend the manuscript for publication after a few issues are considered by the authors, especially to improve the readability for a general readership.

1. I think the title itself "Further insight" refer to a previous paper from the author ("further" with respect to what?) and might be simplified to "Insights on" or "On the role..."

2. There is some literature missing that can be added: for instance
i) the parameter estimation problem (Introduction, lines 1-10) may be approached also with adjoint techniques, and I recommend the authors to mention this alternative methodology;
ii) in the description of twin experiments with perturbations (P4L1-6), there are many analogies with OSSEs (Observing system simulation experiments) that can be mentioned as well.

3. The authors often refer to simple climate/coupled model. I suggest them to always use the definition of "conceptual model" as it can hardly be considered a climate model

4. The reader is too much referred to literature in the Methodology section. For instance, I had to understand only through referred papers
i) the size of the conceptual model of Eq. (1) is never discussed (is it a single-column model or a limited-size model? What are the boundary conditions of the problems, if any?)
ii) little is said about the EAKF, which might be better introduced from a theoretical point of view and in terms of advantages/disadvantages w.r.t. other filters and data assimilation methods. I guess the authors choose it for its ease in the parameter estimation, but this can be better clarified
iii) for such a small size problem, a 20-member ensemble size appears quite small without reason. Clearly the problem size is small, but it is worth mentioning sensitivity tests performed on the ensemble size.

5. I found the conclusion in P7L3.9 on preferring atmospheric to ocean observations to determine ocean parameters very dependent on the conceptual model the authors use. First, some parameters (c2) are not ocean parameters but coupling parameters, strictly speaking; second, the "first guess" of the ocean parameters themselves, determining time scales and interactions, may not necessarily represent the real world; third, the observing network that observe ocean and atmosphere state may be not representative of the real observing networks. I would mention the limits of the conceptual model rather than emphasize this conclusion.

6. Since Section 3 contains a lot of information and experiments, I suggest to add a paragraph between the 1st and 2nd paragraph of Section 4 to summarize some results from the experiments on individual/combined state and parameter estimation.

Language issues

weak coupled → weakly coupled (P4L10 and further occurrences)
P4L21 "And also considering...visualization" sounds very awkward

---

## Referee Comment (RC2) · Anonymous Referee #2 · 11 Nov 2016

Review of "Further Insights on the Role of Accurate State Estimation in Coupled Model Parameter Estimation by a Simple Climate Model Study" by Yu et al.

This work used a simple conceptual model to provide insights on the role of atmospheric/oceanic state estimation in coupled model parameter estimation. They concluded that the accuracy of the atmospheric state is the crucial factor for such kind of parameter estimation. I regard this work is innovative and the manuscript is well structured. However, my main concern is whether the setup of the assimilation experiments and the conclusion of this work are applicable to the real world. My suggestion for this manuscript is major revision before it can be considered for formal publication. My main concerns are as follows.

**Major comments:**

1. The setup of the SE has an assimilation interval of 5 time-steps, which is shorter than the current atmosphere analysis update interval and can be regarded as a rapid update cycle. Such setup also greatly controls the signal-to-noise of the atmospheric condition. Although the authors claim that the results are not sensitive to the choice of update interval (Page 5, line 13), the accuracy of the atmospheric state could be seriously degraded with a longer update interval (or with only $x_1$ observations) and shed the relationship with the parameter.
   - Can the authors provide PE experiments using a longer update interval (e.g. 25 TU) or assimilate $x_1$ only to illustrate the condition that the atmosphere state is less optimally observed?

2. Do the parameter spread and the amount of inflation need to be well tuned? How important are the choices for tuning the parameter spread and the amount of inflation? I suggest that the authors could link the parameter uncertainties to those appear in realistic coupled model, e.g. $a_2$ mimics the heat flux for atmosphere and $c_2$ mimics the windstress for ocean (also see the comment #3).
   - The uncertainties of parameters $a_2$ and $c_2$ (Fig. 2 and Fig. 3) are one-order different. Are they chosen on purpose? What are the averaged ensemble spreads for these two parameters? What happened if one chooses to remain a larger and same amount of uncertainty for these two parameters?
   - If we can provide an unbiased $a_2$, could assimilation w-only lead to a

successful parameter estimation for $c_2$?

- Page 4, Line 25: PE starts 40 TU later than SE. It should be clarified that the purpose is to constrain the accuracy of states (as stated at line13, Page 7). Why is it so important?

- Compared with Fig. 2b and Fig.3b, the ensemble mean in Fig. 2c and Fig. 3c does not locate near the middle of the ensemble distribution after PE converges. Does this mean that the parameter ensemble distribution is skewed? Is there a particular reason for this result?

3. The ensemble spread of the parameter $a_2$ seems to be less than 5% (and will be inflated when the spread is smaller than 0.6%). Is this realistic? In realistic setup of climate modeling, the uncertainties in the parameters associated with air-sea interaction (wind stress, heat flux) could be as large as 10%, in addition to bias in these parameters. The setup of the PE experiments may be too ideal to project the conclusion to realistic coupled modeling. In reality, there are several challenging issues in parameter estimation within atmospheric/ocean assimilation frameworks. However, such real and major obstacles cannot be explained by the results of the simple model.

- In realistic parameter estimation using EnKF, how to construct a reliable error covariance between parameter and observation increments could be still challenging. In this simple model, one can easily perturb the parameter with the white noise without considering the characteristics of the horizontal structure. However, in reality, the structure of the ensemble perturbations of the parameter determines the pattern of the corrections away from the observations and how to keep a reasonable perturbation structure for parameters becomes a challenging task, especially for the parameters used in atmosphere model.

- So far, we may not have enough observation information for parameter estimation or constrain the parameter uncertainty (e.g. surface/near surface atmosphere observations that can reflect the air-sea interaction).

- I suggest the authors could provide some discussion about improving the accuracy of the atmosphere state for parameter estimation in real ocean modeling. What are the current limitations and what can be done?

**Minor suggestions:**

1. I suggest including the bias and root mean square error of the states and parameters in Table 1.

2. Line 5: "tuning" procedure?

3. Page 3, it will be easier for the readers if the authors can give a physical meaning for parameters $a_2$ and $c_2$.

4. Page 6, Line 18: Shouldn't the zigzag shape mainly due to the update from assimilation of observations?

5. I suggest that some paragraphs can be clarified or re-arranged.
   - The first paragraph in Section 3 is somewhat confusing. I suggest starting from Table 1 and explain the differences among the experiments.
   - Is the experiment mentioned for Fig. 2a and Fig.3a (both atmosphere SE and ocean SE) included in Table 1?

---

## Referee Comment (RC3) · Anonymous Referee #3 · 13 Nov 2016

This paper investigates how the model parameter estimation works in an EnKF for an atmosphere-ocean coupled system. This study performs a series of parameter estimation experiments using a low-order, toy system based on the famous Lorenz-63 three-variable model but with an extension of additional near-surface and deep ocean components. The results are somewhat interesting that the fast atmospheric component's state estimation plays a key role in the parameter estimation problem both for the ocean-atmosphere coupling coefficient $c_2$ and the internal dynamical parameter $a_2$ for the second atmospheric variable $x_2$. I find the topic of parameter estimation stability jointly with state estimation stability is very interesting, and this paper is a useful contribution in the field, although could be done better. I find the value of publishing this

article, but I found some issues that need to be addressed before final publication as below:

1. There are a number of grammatical errors, which need to be corrected.

2. "Signal-to-noise" of the ensemble-based error covariance between the states and parameters appears repeatedly, but there is no direct investigation about it. Since this study performs idealized toy-model experiments, I would assume that the authors may find a better way of investigating and presenting the signal-to-noise more explicitly.

3. P.7, L.7-9, "Here our results suggest that in a coupled system, to determine oceanic coefficients, it is more important to get more atmospheric measurements to constrain the atmospheric states than to get more oceanic measurements to directly apply to oceanic PE." This is an interesting hypothesis inspired by the simple toy model results, but this statement seems to be an overgeneralization. The real coupled atmosphere-ocean system is much more complicated than the two-time-scale toy system with only 3 atmospheric and 2 oceanic variables. This statement should be a hypothesis or speculation at this point.

4. P.7, L.21-22, "reducing x2 uncertainty is critical", I do not find this statement well supported or proven by the experimental results. This statement seems to be a hypothesis or speculation.

Minor comments:

1. Eq. (2) does not contain observation error statistics, and I am curious how to interpret this equation intuitively. I understand that this equation gives analysis increments for the ith ensemble member. The analysis increments should balance between the observation error and background error. This equation has only the background error variance in the observation space as the denominator, but does not contain the observation error variance which usually appears in the data assimilation equations as an R matrix.

2. P.6, L.30, eta-to-c6 PE suddenly appears here, without any description about observations for eta (deep ocean state variable). Section 2.2 described only x2 and w observations, and the readers would assume the experiments use only x2 and w observations.
* * *

---

## Author Comment (AC1) · 24 Nov 2016

A few conferences were held for authors to discuss the comments of 3 reviewers. All comments of reviewer 1 have been fully addressed in the revision. The first authors learned a lot from such commented points, and here the authors want to express great appreciation for the advice of the reviewer. The paper is revised as the reviewer's suggestions. New tables have been added in the paper. We also exchanged the location of section 2.2 and 2.3.

See the attachments.

Please also note the supplement to this comment:

[Figure]

http://www.nonlin-processes-geophys-discuss.net/npg-2016-52/npg-2016-52-AC1-supplement.zip

---

## Author Comment (AC2) · 24 Nov 2016

A few conferences were held for authors to discuss the comments of 3 reviewers. All co-authors converged to the point that all comments from reviewer 2 are very thoughtful and important for improving the manuscript and enhancing our understanding on the topic. Several experiments for explaining the concerns of the reviewer are performed. The paper is renewed as the reviewer's suggestions. New tables have been added in the paper. We also exchanged the location of section 2.2 and 2.3.

See the attachments for details. Thanks!

Please also note the supplement to this comment:

[Figure]

http://www.nonlin-processes-geophys-discuss.net/npg-2016-52/npg-2016-52-AC2-supplement.zip
* * *

---

## Author Comment (AC3) · 24 Nov 2016

All authors appreciate greatly for the encouragements and comments from reviewer 3. Coordinating with addressing the comments of reviewers 1 & 2, all points that reviewer 3 concerns have been fully addressed in the revision. The paper is renewed as the reviewer's suggestions. New tables have been added in the paper. We also exchanged the location of section 2.2 and 2.3. See the attachments for details. Thanks!

Please also note the supplement to this comment:
http://www.nonlin-processes-geophys-discuss.net/npg-2016-52/npg-2016-52-AC3-supplement.zip

---

## Author Response (AR1)

Point-by-point responses for review #1

In general:

**A few conferences were held for authors to discuss the comments of 3 reviewers. All comments of reviewer 1 have been fully addressed in the revision. The first authors learned a lot from such commented points, and here the authors want to express great appreciation for the advice of the reviewer. The paper is revised as the reviewer's suggestions. New tables have been added in the paper. We also exchanged the location of section 2.2 and 2.3. What follows is a point-by-point reply for reviewer 1:**

General comment:

The manuscript investigates the impact of observational constraints, through data assimilation methods, on coupled model state and parameter estimation using a conceptual 5-variable model. I found the manuscript interesting and appropriate for the journal, especially to fill the existing gap of idealized studies in coupled data assimilation experiments. It is somehow less relevant, in my opinion, for parameter estimation, given the complexity of real-world CGCM, as the authors themselves discuss in the Conclusions.

I recommend the manuscript for publication after a few issues are considered by the authors, especially to improve the readability for a general readership.

**RE: Thanks for your encouragement. All issues are replied point-by-point as below. We hope the whole manuscript has been essentially improved.**

1. I think the title itself "Further insight" refer to a previous paper from the author ("further" with respect to what?) and might be simplified to "Insights on" or "On the role..."

**RE: The title is changed to "Insights on the Role of Accurate State Estimation in Coupled Model Parameter Estimation by a Conceptual Climate Model Study".**

2. There is some literature missing that can be added: for instance

i) the parameter estimation problem (Introduction, lines 1-10) may be approached also with adjoint techniques, and I recommend the authors to mention this alternative methodology;

**RE: Three different choices for the parameter estimation including the adjoint techniques and related references are added in P2L10~11.**

ii) in the description of twin experiments with perturbations (P4L1-6), there are many analogies with OSSEs (Observing system simulation experiments) that can be mentioned as well.

**RE: Thanks for this suggestion. Our twin experiment is a kind of OSSEs. This and references of other PE under OSSEs are added in P5L11~12.**

3. The authors often refer to simple climate/coupled model. I suggest them to always use the definition of "conceptual model" as it can hardly be considered a climate model

**RE: Yes, done. Thanks.**

4. {The reader is too much referred to literature in the Methodology section. For instance, I had to understand only through referred papers

i) the size of the conceptual model of Eq. (1) is never discussed (is it a single-column model or a limited-size model? What are the boundary conditions of the problems, if any?)}

**RE: Our model is a low-order (limited-size) conceptual model. The boundary condition is a predefined seasonally-varying solar radiation $S(t)=Sm+Ss\ COS(2*pai*t/Spd)$, which is a simple and idealized approximation of the real world boundary condition. New lines of this introduction are added in P3L22~25.**

{ii} little is said about the EAKF, which might be better introduced from a theoretical point of view and in terms of advantages/disadvantages w.r.t. other filters and data assimilation methods. I guess the authors choose it for its ease in the parameter estimation, but this can be better clarified}

**RE: Thanks for the suggestions. More details about the EAKF method including its advantages/disadvantages was added as an independent paragraph on the beginning of section 2.2, P4L1~10.**

{iii) for such a small size problem, a 20-member ensemble size appears quite small without reason. Clearly the problem size is small, but it is worth mentioning sensitivity tests performed on the ensemble size.}

**RE: Thanks for the suggestion. We performed sensitivity tests on the member size. The result is shown in the following figure. Generally speaking, the RMS error of the mean parameter will increase when lowering the ensemble size. But it can also be clearly seen from the panel (d) that no matter how big the ensemble size is, the result with an oceanic SE is unacceptable. We think size 20 is enough for showing the difference between the successful and the failure cases. We added new lines to clarify this problem in P4L24~28 and P7L18~20.**

[Figure]

**Figure caption: Time series of the parameter in 4 test cases with different**

ensemble size settings. SE $x_2$, PE $x_2$ to $a_2$ (abc), SE $w$, PE $w$ to $a_2$ (d). The ensemble size is 5 (a), 10 (b), 40 (c) and 40 (d) respectively.

5. {I found the conclusion in P7L3.9 on preferring atmospheric to ocean observations to determine ocean parameters very dependent on the conceptual model the authors use. First, some parameters (c2) are not ocean parameters but coupling parameters, strictly speaking;}

RE: Thanks for this comment. The $c_2$ is more like a coupling parameter than pure ocean parameter. Therefore we performed experiments about $c_6$ as an complementary (Fig. 5). The necessity of an atmospheric SE still holds.

{second, the "first guess" of the ocean parameters themselves, determining time scales and interactions, may not necessarily represent the real world; }

RE: The conceptual model cannot fully represent the real world. But it has great advantage for clarifying the PE problem without sufficient observations. The parameters are set to simulate the parameterization of CGCM. We added more description and discussion about the simple model parameters in P3L25~28.

{third, the observing network that observe ocean and atmosphere state may be not representative of the real observing networks. I would mention the limits of the conceptual model rather than emphasize this conclusion.}

RE: Thanks for the reviewer's suggestion. The real world observation generally has strong temporal and geographical dependency. The real data are always with all kinds of incomplete. All these flaws motivate us carrying out this partial SE research at the first place. Some of them cannot be represented in our model because this conceptual model does have its limits though its dynamics and transferring of the uncertainty is crystal clear. Following the suggestion, we added new lines to discuss these limits in section 4, P11L19~27.

6. Since Section 3 contains a lot of information and experiments, I suggest to add a paragraph between the 1st and 2nd paragraph of Section 4 to summarize some results from the experiments on individual/combined state and parameter estimation.

**RE: A new paragraph summarizing all experiments and the direct results was added as the reviewer's suggestion in section 4, P11L4~11.**

Language issues

{weak coupled → weakly coupled (P4L10 and further occurrences) }

**RE: Done. Thanks.**

P4L21 "And also considering...visualization" sounds very awkward

**RE: The sentence was rewritten to "Therefore we set the ensemble initial values of $a_2$ as a Gaussian distribution N(30, 1) (30 as the mean value and 1 as the standard deviation), and the spread is enough for the model ensemble uncertainty. The ensemble initial values of $c_2$ are set as N(0.8, 0.5)." as in P5L30~32. Thanks.**

Point-by-point responses for review #2

In general:

**A few conferences were held for authors to discuss the comments of 3 reviewers. All co-authors converged to the point that all comments from reviewer 2 are very thoughtful and important for improving the manuscript and enhancing our understanding on the topic. Several experiments for explaining the concerns of the reviewer are performed. The paper is renewed as the reviewer's suggestions. New tables have been added in the paper. We also exchanged the location of section 2.2 and 2.3. What follows is a point-by-point reply for reviewer 2:**

General comment:

This work used a simple conceptual model to provide insights on the role of atmospheric/oceanic state estimation in coupled model parameter estimation. They concluded that the accuracy of the atmospheric state is the crucial factor for such kind of parameter estimation. I regard this work is innovative and the manuscript is well structured. However, my main concern is whether the setup of the assimilation experiments and the conclusion of this work are applicable to the real world. My suggestion for this manuscript is major revision before it can be considered for formal publication. My main concerns are as follows.

**RE: Thanks for the reviewer's comments. The PE without sufficient SE in a coupled system is an interesting topic. We gain a lot of benefits from this study, for example, the real analysis and prediction with the coupled data assimilation (CDA) system. While our coupled data assimilation (CDA) system was established in 2007, we have been making efforts to implement parameter estimation into CDA to improve climate analysis and prediction, but the improvement remains in a limited range or none. We have to come back to simple models to sort out the sources of noises. The simple conceptual model does have limits (added in section 4, P11L19~27), but its dynamics and transferring of the uncertainty is crystal clear. With the help of this model, we**

found that since the imperfection of observing system and extra model errors have much stronger influences on coupled parameter estimation than coupled state estimation, how to enhance the signal-to-noise ratio of a parameter-state covariance is the key for successful coupled model parameter estimation. In such cases, the simple model has more visibility to demonstrate the essence of the problem.

**Major comments:**

{1. The setup of the SE has an assimilation interval of 5 time-steps, which is shorter than the current atmosphere analysis update interval and can be regarded as a rapid update cycle. Such setup also greatly controls the signal-to-noise of the atmospheric condition. Although the authors claim that the results are not sensitive to the choice of update interval (Page 5, line 13), the accuracy of the atmospheric state could be seriously degraded with a longer update interval (or with only $x_1$ observations) and shed the relationship with the parameter.

Can the authors provide PE experiments using a longer update interval (e.g. 25 TU) or assimilate $x_1$ only to illustrate the condition that the atmosphere state is less optimally observed? }

RE: Thanks for this important comment since the signal-to-noise ratio is different on different frequency in state variability. As suggested, we firstly tested our result with different PE update intervals. The following figures show that the major results in our studies do not depend on the PE interval settings. New lines are added in P7L20~22.

[Figure]

**Figure caption: Time series of the estimated $c_2$ ensemble in the $w$-to-$c_2$ PE experiment with SE for $w$ only, when the PE update interval is 0.02 TU (i.e. 2 time steps) (upper-left), 0.05 TU (upper-right), 0.25 TU (lower-left) and 2.5 TU (lower-right).**

{2. Do the parameter spread and the amount of inflation need to be well tuned? How important are the choices for tuning the parameter spread and the amount of inflation? I suggest that the authors could link the parameter uncertainties to those appear in realistic coupled model, e.g. $a_2$ mimics the heat flux for atmosphere and $c_2$ mimics the windstress for ocean (also see the comment #3). }

**RE: There are two considerations referring to the parameter spread and the amount of inflation. Firstly, as shown in the following figure, a smaller inflation level will enlarge period of the fluctuation of the black thick line. If the fluctuation period of the mean parameters is too long, then the effect is somehow similar to a slower convergence rate of the mean result during an arbitrary diagnostic window. Secondly, a too large inflation will cause the spread jump out**

of the reasonable range. Within a relatively large scope, the inflation level will only change the spread of the parameter, but not change the mean of its ensemble. Because the mean value of the parameter is not as sensitive as the spread to the inflation level. It would not affect our main results in this paper.

[Figure]

Figure caption: Time series of the estimated $a_2$ ensemble in the $x_2$-to-$a_2$ PE experiment with SE for $x_2$ only, the limited inflation value is 0.01(a), 0.05(b), 0.2 (c), 0.4(d), 0.6(e), 1.0(f). 0.2 is the value used in the paper.

{The uncertainties of parameters $a_2$ and $c_2$ (Fig. 2 and Fig. 3) are one-order different. Are they chosen on purpose? What are the averaged ensemble spreads for these two parameters? What happened if one chooses to remain a larger and same amount of uncertainty for these two parameters? }

RE: Thanks for the reviewer's thoughtful suggestion. We tried to test different initial bias combination at first, but sooner we found that in the original system, the value of $a_2$ is too much limited by the chaotic nature of the Lorenz equation (the $a_2$ can not be perturbed too much or the Lorenz equation otherwise will loose its chaotic nature and makes the experiment fail). We tried to avoid this by changing the system from two-way coupling to one-way coupling in section 3, see Appendix A. Although $a_2$ still cannot be changed too much, $c_2$ and $c_6$ can be changed in a wide scope. In such a circumstance, $c_6$ interacts with $w$ and $\eta$, both being strongly forced by the periodic cosine function more than the Lorenz chaotic forcing. On the contrary, unlike $c_6$, no matter how periodic $w$ is, $c_2$ is

always affected by the chaotic $x_2$. The experiments with varying $S_s$ values give a lot of insights on this issue.

{If we can provide an unbiased $a_2$, could assimilation w-only lead to a successful parameter estimation for $c_2$?}

RE: For all of the experiments, only the parameter being estimated is biased from its truth. In experiment $w$-to-$c_2$, the $a_2$ is unbiased. And From Table 1, it clearly shows that even with an unbiased $a_2$, assimilation $w$-only will not lead to a successful parameter estimation for $c_2$. As in equation (1), the state variable $w$ is calculated from $c_2$ and $x_2$. The $x_2$ is chaotic even with an unbiased $a_2$, therefore, the correlation between $c_2$ and $w$ is disturbed by the chaotic $x_2$, and the correlation is not helpful during the estimation of the value of $c_2$ from the difference between $w$ and "$w$ observation."

{Page 4, Line 25: PE starts 40 TU later than SE. It should be clarified that the purpose is to constrain the accuracy of states (as stated at line13, Page 7). Why is it so important? }

RE: Yes, more discussions and justifications are added in the revision. Please see P6L1-4, P7L7-10.

{Compared with Fig. 2b and Fig.3b, the ensemble mean in Fig. 2c and Fig. 3c does not locate near the middle of the ensemble distribution after PE converges. Does this mean that the parameter ensemble distribution is skewed? Is there a particular reason for this result? }

RE: The thick black line indicates the ensemble mean of the parameter. As in Fig. 2b, 3b, 2c, 3c, all the thick black lines are near the referenced thin line enough to be called a successful PE. The difference between Fig. c and b is mainly about the asymmetry of the spread (shaded area). The asymmetry suggests that in the fully SE experiments (Fig. 2c, 3c) the distributions of the 20 ensemble members are not very Gaussian like. This actually exhibits advantage of the EAKF method

**that it can "adjust" and keep the distribution of the ensembles. More description about EAKF are added in P4L1~10.**

{3. The ensemble spread of the parameter $a_2$ seems to be less than 5% (and will be inflated when the spread is smaller than 0.6%). Is this realistic? In realistic setup of climate modeling, the uncertainties in the parameters associated with air-sea interaction (wind stress, heat flux) could be as large as 10%, in addition to bias in these parameters. The setup of the PE experiments may be too ideal to project the conclusion to realistic coupled modeling. In reality, there are several challenging issues in parameter estimation within atmospheric/ocean assimilation frameworks. However, such real and major obstacles cannot be explained by the results of the simple model.}

**RE: The value of $a_2$ is too much limited by the chaotic nature of the Lorenz equation. But with the one-way coupling, we tried experiments with huge state fluctuations (different sets of $S_s$), the change of the forcing states is even bigger than in the real world. The main result of our paper still holds.**

{In realistic parameter estimation using EnKF, how to construct a reliable error covariance between parameter and observation increments could be still challenging. In this simple model, one can easily perturb the parameter with the white noise without considering the characteristics of the horizontal structure. However, in reality, the structure of the ensemble perturbations of the parameter determines the pattern of the corrections away from the observations and how to keep a reasonable perturbation structure for parameters becomes a challenging task, especially for the parameters used in atmosphere model. }

**RE: Thanks for the reviewer's comments. In this simple model, the construction of a reliable error covariance is indeed easier than in the real world. It is mainly because the observation is perfectly consistent to the model dynamics. In the real world, the structure is greatly geophysical dependent. The study is considered to be the first stepping-stone for further studies with more complex models. Therefore, we added new paragraph to fully discuss the limitation of our work in section 4, P11L19~27.**

{So far, we may not have enough observation information for parameter estimation or constrain the parameter uncertainty (e.g. surface/near surface atmosphere observations that can reflect the air-sea interaction). I suggest the authors could provide some discussion about improving the accuracy of the atmosphere state for parameter estimation in real ocean modeling. What are the current limitations and what can be done? }

**RE: As the parameter estimation in real ocean modeling can be very geophysical dependent, with our conceptual model, two things are suggested to be important for the further studies. The first one is our studies suggest that PE of oceanic parameter is possible to succeed with only the atmospheric observations. Considering there are also regions where the coupling effect is weak, adaptive measurements for different region seem important and necessary. Another suggestion is that the PE technique can be improved to perform separately at multiple-scales. All these require further research work to clarify. The discussion is rewritten in section 4, P11L28~P12L4.**

**Minor suggestions:**

1.  I suggest including the bias and root mean square error of the states and parameters in Table 1.

**RE: Thanks for the reviewer's suggestion. A new table 2 was added in the revision showing the root mean square bias error of the state variable and the parameter during the last 100 TUs in 8 PE experiments.**

2. Line 5: "tuning" procedure?

   **RE: The sentence is rewritten.**

3. Page 3, it will be easier for the readers if the authors can give a physical meaning for parameters $a_2$ and $c_2$.

   **RE: Several lines elaborate the physical meaning of the two parameters are added in section 2.1, P3L25~28.**

4. Page 6, Line 18: Shouldn't the zigzag shape mainly due to the update from assimilation of observations?

   **RE: The zigzag shape is mainly due to the inflation process. The spread of the ensemble member is continuously shrinking during the PE process. After a while, when the std (spread) of the parameter ensemble is below some limit (40% of its initial spread), we inflate the ensemble by multiply a constant factor to the parameter anomalies to satisfy this STD value. More accurate description is added in P5L2.**

   **Sometimes the constant mag factor will be used several times for the spread to go beyond the limit. The mean value of the parameter is not sensitive to this factor. After the multiplication, the spread will generally be higher than the limited value (form the zigzag shape) to make sure the inflation would not immediately happen again.**

5. I suggest that some paragraphs can be clarified or re-arranged. The first paragraph in Section 3 is somewhat confusing. I suggest starting from Table 1 and explain the differences among the experiments. Is the experiment

mentioned for Fig. 2a and Fig.3a (both atmosphere SE and ocean SE) included in Table 1?

**RE: Thanks for the suggestion. The first paragraph in Section 3 (P6L6~14) has been rewritten. The experiments of Fig. 2a and Fig. 3a are not included in Table 1. The main purpose of this paper is to discuss the 8 PE experiments with partial SE. We shown the coupled SE experiment as a standard reference level for the partial SE cases. They are not very relevant to our main purpose.**

Point-by-point responses for review #3

In general:

**All authors appreciate greatly for the encouragements and comments from reviewer 3. Coordinating with addressing the comments of reviewers 1 & 2, all points that reviewer 3 concerns have been fully addressed in the revision. The paper is renewed as the reviewer's suggestions. New tables have been added in the paper. We also exchanged the location of section 2.2 and 2.3. What follows is a point-by-point reply for reviewer 3:**

General comment:

This paper investigates how the model parameter estimation works in an EnKF for an atmosphere-ocean coupled system. This study performs a series of parameter estimation experiments using a low-order, toy system based on the famous Lorenz-63 three-variable model but with an extension of additional near-surface and deep ocean components. The results are somewhat interesting that the fast atmospheric component's state estimation plays a key role in the parameter estimation problem both for the ocean-atmosphere coupling coefficient c2 and the internal dynamical parameter a2 for the second atmospheric variable x2. I find the topic of parameter estimation stability jointly with state estimation stability is very interesting, and this paper is a useful contribution in the field, although could be done better. I find the value of publishing this article, but I found some issues that need to be addressed before final publication as below:

**RE: Thanks for your encouragement. All issues are replied point-by-point as below. We hope the whole manuscript has been essentially improved.**

1. There are a number of grammatical errors, which need to be corrected.

**RE: A few rounds of reading/editing from native English speaker were conducted. The grammatical errors were fixed. Thanks.**

2. {"Signal-to-noise" of the ensemble-based error covariance between the states and parameters appears repeatedly, but there is no direct investigation about it. Since this study performs idealized toy-model experiments, I would assume that the authors may find a better way of investigating and presenting the signal-to-noise more explicitly.}

**RE: Thanks for the suggestion. The signal-to-noise ratio of the ensemble-based error covariance between the states and parameters is better to be diagnosed in SE only experiments. In these runs, there are no PE processes to fix the biased parameter spread so that parameter perturbations can be fully transferred to the model states. Then the state-parameter covariance can be checked without any disturbance from a PE correction. Following the previous work of Zhang et al. (2012), we defined a new index (called $r_{s2n}$) to measure the signal-to-noise ratio of the ensemble-based error covariance between the states and parameters. The best (worst) representation of the signal-to-noise ratio is characterized by a $r_{s2n}$ value of 1(0). A new Table 3 is added in the paper. It shows all $r_{s2n}$ and related values in the 8 SE only (no PE) experiments. Description of this index and related discuss is added in P9L16~27.**

3. {P.7, L.7-9, "Here our results suggest that in a coupled system, to determine oceanic coefficients, it is more important to get more atmospheric measurements to constrain the atmospheric states than to get more oceanic measurements to directly apply to oceanic PE." This is an interesting hypothesis inspired by the simple toy model results, but this statement seems to be an overgeneralization. The real coupled atmosphere ocean system is much more complicated than the two-time-scale toy system with only 3 atmospheric and 2 oceanic variables. This statement should be a hypothesis or speculation at this point.}

**RE: The sentences are rewritten as in P8L10~12. Thanks.**

4. { 4. P.7, L.21-22, "reducing x2 uncertainty is critical", I do not find this statement well supported or proven by the experimental results. This statement seems to be a hypothesis or speculation.}

**RE: The statement was changed to "Instead, reducing x2 uncertainty (enhancing the estimation accuracy of the atmospheric states) is more relevant to the solution of the problem." (P8L24~25)**

Minor comments:

1. {Eq. (2) does not contain observation error statistics, and I am curious how to interpret this equation intuitively. I understand that this equation gives analysis increments for the ith ensemble member. The analysis increments should balance between the observation error and background error. This equation has only the background error variance in the observation space as the denominator, but does not contain the observation error variance which usually appears in the data assimilation equations as an R matrix.}

**RE: The observation error variance is calculated before this projection process. The observation are firstly compared to their simulated values, the difference between them are manipulated to produce the observational increments. The production of the observational increments considers the observation error variance and its PDF. The observation error is set as a constant number in our simulation. The standard deviation of "observational" errors are 2 for the atmospheric variables $x_{1,2,3}$ and 0.2 for the oceanic variable $w$. New introduction of the EAKF method is added in the section 2.2, P4L1~10.**

2. P.6, L.30, eta-to-c6 PE suddenly appears here, without any description about observations for eta (deep ocean state variable). Section 2.2 described only x2 and w observations, and the readers would assume the experiments use only x2 and w observations.

**RE: Thanks for the reviewer's suggestion. Description about observations for eta is added in the new $\eta$-to-$c_6$ section (P7L31~34). It directly points out that the experiments uses $\eta$ only for the PE and uses all state variables for the SE.**

[revised manuscript text omitted]

---

## Referee Report (RR1)

**Second review of "Insights on the Role of Accurate State Estimation in Coupled Model Parameter Estimation by a Conceptual Climate Model Study" by X. Yu et al.**

I am satisfied with all changes made by the authors, which address all my previous comments. In particular, I appreciate the inclusion of the discussion on the limits of applicability of the study added in the Conclusion and Discussion Section and due to the observing network and neglecting of chaotic ocean processes.

The only point is that the size of the problem still remains mysterious, and I suggest the authors to mention it, as the impact of the ensemble size clearly depends on the problem size, and this is not mentioned.

---

## Referee Report (RR2)

The manuscript addresses an important topic of parameter estimation in coupled atmosphere-ocean climate models. Performance of Ensemble Kalman Filter approach for joint estimation of model state and parameters is assessed in twin data assimilation experiments with a "conceptual climate model". The conceptual model is a system of five first-order ordinary differential equations describing evolution of three "atmospheric" and two "oceanic" state variables. The results of these experiments allow the authors to conclude that "*enhancing estimation accuracy of atmospheric states is very important for the success of coupled model parameter estimation, especially for the parameters in the air-sea interaction processes*". I believe that the test of the technique within the frames of oversimplified model and synthetic observations is a necessary first step in the development of robust and efficient methods for parameter estimation in the modern climate models. The results of such idealized study and the authors' experience in the design of the parameter estimation experiments could be of interest for data assimilation community and is worth publishing. On the other hand I cannot recommend publishing the manuscript in its present form for several reasons listed below.

**Major comments**

**Comment 1.** I found the manuscript to be difficult to read. It is not because of grammar errors as in the first version of the manuscript. Some parts of the revised manuscript lack clarity and logic in presenting the material. I had to read some sentences and paragraphs several times trying to understand what the authors intended to say. As an example I will cite here just one paragraph in section 2.3 (P3L16):

"*The different PE experiments can be distinguished in 3 perspectives: 1) 2 state constraint settings (i.e. SE settings) that assimilate the atmosphere "observations" (x2) only, and the ocean "observations" (w) only; 2) 2 parameter settings – a2 in the atmosphere equation, and c2 in the ocean equation; 3) 2 observational settings – one atmosphere (x2), and one ocean (w). Here the SE uses weakly coupled data assimilation as termed in the literature (e.g. Lu et al., 2015), i.e., x2 observations impact on all x variables, and w (η if applicable) observations impact on w (η) itself (considering the different time scales of w and η, no cross-impact between them), while the PE could use different medium observations. Therefore, eventually we have a few PE cases with full SE – both x and w are constrained by their observations, and particularly 8 PE cases with partial SE – only some medium is constrained by its observations. These PE cases have different SE accuracy. Through thoroughly analysing these PE cases, we are able to detect the influence of the SE accuracy in different medium on coupled model PE*".

In the first sentence of this paragraph it is difficult to see the difference between "perspectives" 1) and 3). The second sentence should probably be more appropriate in the section 2.2 describing the ensemble filter. This sentence implies essential modification (simplification) of the algorithm which probably deserves a more detailed discussion in the section 2.2.
The second sentence contradicts the first one since the first sentence mentions only "2 observational settings – one atmosphere (x2), and one ocean (w)" while the second sentence assumes assimilation of η observations "if applicable".
The third sentence ignores assimilation of η again. The reader has to guess what are the "few PE cases with full SE" and why there are exactly 8 PE cases with partial SE. If assimilation of η observations is disregarded, with "2 observational settings" and "2 parameter settings" the reader will not be able to get 8 PE cases with partial SE.

**In my view, the authors should perform a significant work on "polishing" the manuscript to improve clarity of the material presentation.**

**Comment 2.** As with any suboptimal data assimilation method, the results of state and parameter estimation presented in the manuscript depend on particular realization of the method. That is why it is important to provide a clear description of the method what is sufficiently detailed to allow other

researches to repeat the calculations. I believe that the section 2.2 falls short of these expectations. It seems that at least the Reviewer 3 of the previous version of the manuscript did not get clear understanding of the method either. The Reviewer 3 wrote in his comments:

"*Eq. (2) does not contain observation error statistics, and I am curious how to interpret this equation intuitively. I understand that this equation gives analysis increments for the ith ensemble member. The analysis increments should balance between the observation error and background error. This equation has only the background error variance in the observation space as the denominator, but does not contain the observation error variance which usually appears in the data assimilation equations as an R matrix*".

In my view, the authors' reply to this comment is disappointing and the revisions to the section 2.2 did make the method description even vaguer.  The authors' reply to the Reviewer 3 comment states:

"*The observation error variance is calculated before this projection process. The observation are firstly compared to their simulated values, the difference between them are manipulated to produce the observational increments. The production of the observational increments considers the observation error variance and its PDF. The observation error is set as a constant number in our simulation. The standard deviation of "observational" errors are 2 for the atmospheric variables x1,2,3 and 0.2 for the oceanic variable w. New introduction of the EAKF method is added in the section 2.2, P4L1~10*".

The following statements are at least inaccurate in this reply:

"*The observation error variance is calculated before this projection process*" – I believe that  the data error variance is set, not computed, since the authors state: "*The standard deviation of "observational" errors are 2 for the atmospheric variables x1,2,3 and 0.2 for the oceanic variable w*".

"*…The difference between*" observations and their simulated values is "*manipulated to produce the observational increments*". What is this "manipulation"?  There is no description of the procedure of this "manipulation" in the revised or original versions of the manuscript.

"*The observation error is set as a constant number in our simulation*" – is it really so? Isn't the observational error a random variable?

The revised version in the section 2.2 is also quite confusing. P4L4 states:

The method "*combines observational probability distribution function (PDF) with model PDF but under an adjustment idea. The algorithm can be sequentially implemented in a two-step procedure (Anderson, 2003): step 1 uses two Gaussian convolution to compute the observational increment at the observational location…*"

First of all, two convolutions of what?  Conventionally convolution is an operation on two functions resulting in another function.  Do the authors mean the convolution of two Gaussian PDFs? But the observational increment is not a function, so it cannot result from convolution of two PDFs.
Secondly, what is the "observational probability distribution function"? Do the authors mean PDF of data error? That is not the same as the PDF of observations. What is the "model PDF"?  Model is not a random variable, it is an operator. Do the authors mean PDF of model forecast error? Do the authors know the PDF of the model forecast error to combine it with observational probability distribution function under an adjustment idea?

Trying to find the answer to the question of the Reviewer 3 I came to the following interpretation of formula (2) in the manuscript:

The authors seem to use well-known stochastic type of ensemble Kalman filter (see e.g. Hamil, 2006) where each ensemble member $x_i$ (model state and parameters) is updated to fit somewhat different realization of observations $y_i$ at the analysis time:

$\Delta x_i = K \Delta y_i$;   $\Delta y_i = H(x_i) - y_i$, $i=1,\ldots,m$

where $y_i = y + e_i$, and $e_i \sim N(0,R)$ is a realization of Gaussian random variable "observational error" with variance (covariance) R. Realizations $e_i$ are different for each ensemble member. The expression for the Kalman gain matrix K in formula (2) is an ensemble estimate for the case of a single observation of the conventional Kalman gain matrix given by the expression:

$K = PH^T (HPH^T + R)^{-1}$

For example, $std(\Delta y)^2$ in the denominator in formula (2) approximates $HPH^T$-R:

$std(\Delta y)^2 = <(\Delta y_i)^2> = < (H(x_i) - y_i) (H(x_i) - y_i)^T>$ ,

where $< >$ denotes ensemble averaging. Under the assumptions that the model is unbiased and e and x are uncorrelated, the expression above can be transformed to

$<(Hx'_i - e_i) (Hx'_i - e_i)^T> = H<x'_i x'_i{}^T>H^T + <e_i e_i{}^T>$

while $<x'_i x'_i{}^T>$ and $<e_i e_i{}^T>$ tend to P and R respectively as the dimension of the ensemble increases.

If my guess is wrong and the the analysis for all ensemble members is performed using the same realization of observations, the data errors are effectively neglected by the filter given by equation (2).

If my guess regarding the meaning of the formula (2) is correct, the authors should describe the procedure of generation of the ensemble of observations in section 2.2. This is an essential part of the method (the method cannot use unperturbed observations). Unfortunately, the authors discuss the observations only in section 2.4 (P5L9):

"*The output of last 103 TUs is then used as the "truth" to produce "observations" by superimposing a white noise on the "observed" variables. The standard deviation of "observational" errors are 2 for the atmospheric variables x1,2,3 and 0.2 for the oceanic variable w.*"

Note that this statement does not specify that the statistics of observational noise is Gaussian. The term "white noise" probably assumes no time correlations in observational errors which is always taken for granted in applications of the Kalman filter. Also it is not clear if the authors "produce" only one realization of observations by superimposing a white noise and "truth" or they produce an ensemble of perturbed observations.

**In my view, the authors should revise the description of the data assimilation method considerably. The present description of the Ensemble Kalman filter method misses some important details which are critical for understanding the data assimilation results.**

**Comment 3.** A number of inconsistencies can be found also in the description of the data assimilation results. For example, the authors mention several times that only the values of x2, w, and "*(η if applicable)*" are perturbed with noise and used as observations. The formula (2) assumes assimilation of scalar (single) data at the analysis time with separate data assimilation for "atmospheric" and "oceanic" sub-models. Conversely, in the statement:

P5L9 "*The standard deviation of "observational" errors are 2 for the atmospheric variables x1,2,3 and 0.2 for the oceanic variable w.*"

the authors specify standard deviations for observational errors in x1 and x3 as if these variables were also used as observations (and do not provide standard deviation for η). Caption to Figure 1 also declares

assimilation of x1,2,3 data: "*a) both the atmosphere (x1,2,3) and ocean (w) from their observations (x1, 2, 3 and w)*".

Another inconsistency in the description of the results is related to the authors' discussion of the time scales of variability of "*fast-varying variables of the atmosphere*" (P3L13) and "*the low-frequency variables of the ocean*" (P3L13).
The authors state that "*the time scale of the w is nearly 10 times of the time scale of the x2*" (P3L13), and that the "*conceptual model mimics very fundamental natures of interactions of three typical time scales in the real world: synoptic (chaotic) atmosphere, seasonal interannual upper tropical oceans and decadal/multi-decadal deep ocean*". This time scale separation was used by the authors as an argument for utilization of a simplified "*weakly coupled data assimilation*" approach (P5L20):
"*x2 observations impact on all x variables, and w (η if applicable) observations impact on w (η) itself (considering the different time scales of w and η, no cross-impact between them)*".
The time scale separation is addressed several times in the interpretation of the results and is used as an explanation of the performance of different parameter estimation experiments, for example: "*The energy of x2 is in the high frequency band and the energy of w is in the low frequency band. x2 varies fast and represents the most uncertain mode, transferrable to low frequency w through the "air-sea" interaction*"(P8L19).

In the first reading of the manuscript, the description of the experiments and analysis of the results convinced me that the atmospheric variable x2 has a well determined spike of energy at periods of few TU, the variable w reacts to coupling with chaotic x2, but is mainly driven by the periodical forcing at the period of 10TU, while η has even longer scales of variability. The understanding that this impression is completely wrong came to me only when I read Appendix 2 there the authors describe an application of the **high-pass filter** to time series of η observations and states to remove "*chaotic signal*" and improve parameter estimation. The authors write:

P12L30 "*... the periodic signal produced by the cosine function has a period of 10TUs (1000 time steps) (defined by Spd in Eq. (1), also see Fig. 10) and the **chaotic signal is much slower** than the periodic signal*".

Since "*chaotic signal*" comes only from atmospheric variables, I was puzzled how could it be "slow". I had to read the results of the experiments with one-way coupled model presented in the section 3.2 several times to find this out. The turning point was Figure 10 showing time evolution of η in the experiments where the amplitude of the periodical forcing was increased 100 to 1000 times compared to the fully coupled model experiments. This figure clearly shows that influence of stochastic atmosphere on oceanic variables is not confined to the periods of few TUs, but dominates the whole range of resolved periods in time series of η. Chaotic atmospheric signal clearly larger than periodic signal in Figure 10 for amplitude of periodical forcing Ss=100, while in the fully coupled model Ss =1. Also, the authors mention that:

P10L14: "*Comparing Fig. 9a to Fig. 6b, it can be seen that the chaotic signal in the one-way coupling model is much smaller than in the original two-way coupling model…*"

My suppositions are that:
a) for the experiments with fully coupled model (amplitude of the periodical force Ss=1) the ocean variables are driven almost entirely by the stochastic atmosphere while the periodical forcing in the equation for w seems to be negligible.
b) the influence of stochastic atmosphere on oceanic variables is not confined to the periods of few TUs, but dominates the whole range of resolved periods in the ocean.
c) time scale separation does not exist and cannot be used as a validation of a simplified "*weakly coupled data assimilation*" approach.
In the light of these suppositions, the authors' conclusions that:

"*enhancing estimation accuracy of atmospheric states is very important for the success of coupled model parameter estimation, especially for the parameters in the air-sea interaction processes*"
or
P8L9: "*This seems different from our previous intuition that in-situ ocean data are always considered as the first important piece of information for determining the oceanic coefficients. Our results here strongly suggest that in the future real coupled model PE experiments, for determining the best coefficient values, no matter the atmospheric or oceanic, sufficient and accurate atmospheric measurements are crucially important*".

looks rather as the consequence of the particular settings of the "conceptual model" parameters and utilization of "*weakly coupled data assimilation*" approach, than as a general feature of parameter estimation that can be extrapolated to realistic climate models.

Finally, note that the amplitude of the periodical force is stated to be 1 in the section 2.1:
P3L18: "($c_1$,     $c_3$,     $c_4$,     $S_m$,   $S_s$,   $S_{pd}$,   $\Gamma$,   $c_5$,   $c_6$) are
            ($10_{-1}$,  $10_{-2}$,  $10_{-2}$,  10,   1,   10,   $10_2$,  1,   $10_{-3}$)"
while in section 3.2 the authors state that $S_s$=10:
P10L14: "*Comparing Fig. 9a to Fig. 6b, it can be seen that the chaotic signal in the one-way coupling model is much smaller than in the original two-way coupling model (with an identical $S_s$ value of 10)*".

**In my view, the authors should revise the description of the model and the sections presenting the results of data assimilation experiments considerably (and possibly redo the experiments to change the relative role of the periodical forcing). The present description may mislead the reader in the same way as it happened to me in my first reading of the manuscript.**

**Minor remarks**

1. Some terms used in the manuscript seem to be misleading and/or not well defined.

These terms are:

a) "*signal-to-noise ratio of error covariance between the model state and parameter*".  The term signal-to-noise ratio is commonly introduced for time series of state variables, but not for covariances.  I agree that it can be used for time series of covariances but that may mislead the reader. Why the authors do not use R shown in Figures 7 and 8 to analyze "relationship between the states and the parameters" quantitatively (I believe R in Figure 7 and 8 is R2= 1 - ||Hx-y||2/||y-mean(y)||2 which is conventionally called the coefficient of determination of linear regression or Least Squares fit). It is possible to call R the time-mean "correlation coefficient". I do not see the reason to multiply R with Sf/Sp and call it the "signal-to-noise ratio", especially because it is not shown in the manuscript that rs2n is a better indicator of parameter estimation performance. Also, in more realistic applications, it might be impossible to assess the ratio Sf/Sp since the full set of observations is not available.

Separation of the signal and noise assumes that where is a model for noise or signal. Here this model is a linear relationship between state variable anomalies and parameter anomalies. I do not see why state variable anomalies and parameter anomalies should exhibit linear relationship at a particular time step. The authors' explanation of this linear relationship is a bit confusing:

P8L26: "*PE completely relies on the covariance between the parameter and model states for projecting the observational information of states onto the parameter. While the PE projection is carried out by a linear regression equation based on the state-parameter covariance (EnKF/EAKF, for instance), only a linear or quasilinear*

*relationship between parameters and states in ensemble is recognized. A hypothesis for all the failed cases without direct atmospheric SE could be that, under such a circumstance, the chaotic disturbances in the atmosphere (Lorenz equations in this case) continuously interacting with the parameter make difficulties for the system to build up a quasi-linear relationship between the state variable and the parameter."*

I agree completely with the first sentence. The second sentence is misleading because the results of linear regression shown in Figures 7 and 8 do not correspond to formula (2). But I agree that "*only a linear or quasilinear relationship between parameters and states in ensemble is recognized*". I do not like the last sentence since no "interaction" between parameters and state is possible. Model states do not act on parameters. Instead of the last sentence I would suggest the following explanation:

PE completely relies on the covariance between the parameter and model states for projecting the observational information of states onto the parameter. The ensemble gets the information on parameter-state covariance entirely from dynamics (not from data): ensemble members obtained in the model forecast with different values of parameters become different with time. The ensemble of states accumulates this covariance information gradually, over time periods which are significantly longer than the data assimilation window of 0.05TUs. Partial assimilation of ocean data effectively restarts the model at every analysis step since oceanic variables are driven by x2 which is not controlled by observations (see Comment 3). As a result, almost no "dynamical" information is accumulated in ensemble covariance. Alternatively, in full data assimilation or in assimilation of x2 data, state corrections at the analysis step are more gentle and some dynamical information on state-parameter covariance has a chance to accumulate with time. And it is not that important if the relationship between parameters and states at a given time step is linear, non-linear or in general there is no relationship at all. It is important that ensemble estimates properly dynamically induced state-parameter covariance.

b) "*state-parameter covariance uncertainties*"- Use of this term assumes that covariance is a random variable that has a "true", mean, or the most probable value. Then "uncertainties" are assessed as some measure of the higher moments of this random variable. This is not discussed in the manuscript.

c) "*chaotic-to-periodic ratio*"- The term itself is fine, but no expression to compute this ratio is given. No numerical values of this ratio for different experiments are presented in the manuscript. It is only mentioned how the authors change this ratio:

P10L11: "*Then we define a chaotic-to-periodic ratio (CPR) in the signals of w ($\eta$) by manipulating the coefficient $S_s$. Eight experiments are performed here, four for w-to-$c_2$ PE and four for $\eta$-to-$c_6$ PE. Each experiment has a different $S_s$ value of 100, 250, 500 and 1000 and thus a reducing CPR in w and $\eta$*".

It is worth to note that by changing the amplitude of $S_s$ the authors also change RMS variability of w and $\eta$ but do not change variance of corresponding observational error. The observations may become much more "accurate" with increase of $S_s$ by the factor of 1000.

Due to vague definition of *signal-to-noise ratio* and *chaotic-to-periodic ratio* the authors mix these two terms. In the Appendix 2 *signal-to-noise ratio* is used instead of *chaotic-to-periodic ratio*.

Also please correct the typo: P12L25 "signal-to-ratio ratio".

3. P3 Eq(1): one of the coefficients Om, Gamma, or Od is redundant.

4. P4L26: "*Thus a practical ensemble size of 20 (applicable for a CGCM) is chosen as a basic experiment setting*". No comparison with CGCM is possible. In the presented research the ensemble size exceeds the dimensions of both the state vector and data. This is not the case for CGCM.

5. P4L30: "as 0.05 TU (i.e. 5 time steps)".  Notation "TU" is not defined on page 4. It is introduced only on page 5.

6. P5L31: "*The ensemble initial values of $c_2$ are set as N(0.8, 0.5)*". What if a particular realization of c2 is negative? The authors state that c2 is "analogous to the drag coefficient $c_d$" which must be positive.

7. P7L20: "*We also performed the experiments under different update interval settings. Test results show that for the issue we are addressing, the conclusion is not sensitive to the update interval if it is within a reasonable range.*" The range of the tested update intervals should be given here.

---

## Author Response (AR2)

[revised manuscript text omitted]

**Point-by-point replies for review #1**

**In general:**

**The comment has been fully addressed in the revision. The following is our reply for reviewer 1:**

I am satisfied with all changes made by the authors, which address all my previous comments. In particular, I appreciate the inclusion of the discussion on the limits of applicability of the study added in the Conclusion and Discussion Section and due to the observing network and neglecting of chaotic ocean processes.

The only point is that the size of the problem still remains mysterious, and I suggest the authors to mention it, as the impact of the ensemble size clearly depends on the problem size, and this is not mentioned.

**RE: For our model, the state vector and data is much smaller than the CGCM, therefore the ensemble size exceeds the size of the states. Following the suggestion, we mention this on P8L4-7 in the revision.**

Point-by-point replies for review #3

In general:

**A few conferences were held for authors to discuss the comments of 3 reviewers. All authors think that all the comments from reviewer 3 are helpful to improve the manuscript and express great appreciation for the advice of the reviewer. Therefore all comments have been fully addressed in the revision. The paper is revised as the reviewer's suggestions. New equation is introduced to explain the method more thoroughly. All mentioned inconsistency is fixed. What follows is a point-by-point reply for reviewer 3:**

The manuscript addresses an important topic of parameter estimation in coupled atmosphere-ocean climate models. Performance of Ensemble Kalman Filter approach for joint estimation of model state and parameters is assessed in twin data assimilation experiments with a "conceptual climate model". The conceptual model is a system of five first-order ordinary differential equations describing evolution of three "atmospheric" and two "oceanic" state variables. The results of these experiments allow the authors to conclude that "enhancing estimation accuracy of atmospheric states is very important for the success of coupled model parameter estimation, especially for the parameters in the air-sea interaction processes". I believe that the test of the technique within the frames of oversimplified model and synthetic observations is a necessary first step in the development of robust and efficient methods for parameter estimation in the modern climate models. The results of such idealized study and the authors' experience in the design of the parameter estimation experiments could be of interest for data assimilation community and is worth publishing. On the other hand I cannot recommend publishing the manuscript in its present form for several reasons listed below.

**RE: Thanks for the reviewer's comments. All issues are replied point-by-point below. Hopefully the whole manuscript has been essentially improved.**

Comment 1. I found the manuscript to be difficult to read. It is not because of grammar errors as in the first version of the manuscript. Some parts of the revised manuscript lack clarity and logic in presenting the material. I had to read some sentences and paragraphs several times trying to understand what the authors intended to say. As an example I will cite here just one paragraph in section 2.3 (P3L16):

"The different PE experiments can be distinguished in 3 perspectives: 1) 2 state constraint settings (i.e. SE settings) that assimilate the atmosphere "observations" ($x2$) only, and the ocean "observations" ($w$) only; 2) 2 parameter settings – $a2$ in the atmosphere equation, and $c2$ in the ocean equation; 3) 2 observational settings – one atmosphere ($x2$), and one ocean ($w$). Here the SE uses weakly coupled data assimilation as termed in the literature (e.g. Lu et al., 2015), i.e., $x2$ observations impact on all $x$ variables, and $w$ ($\eta$ if applicable) observations impact on $w$ ($\eta$) itself (considering the different time scales of $w$ and $\eta$, no cross-impact between them), while the PE could use different medium observations. Therefore, eventually we have a few PE cases with full SE – both $x$ and $w$ are constrained by their observations, and particularly 8 PE cases with

partial SE – only some medium is constrained by its observations. These PE cases have different SE accuracy. Through thoroughly analysing these PE cases, we are able to detect the influence of the SE accuracy in different medium on coupled model PE".

In the first sentence of this paragraph it is difficult to see the difference between "perspectives" 1) and 3). The second sentence should probably be more appropriate in the section 2.2 describing the ensemble filter. This sentence implies essential modification (simplification) of the algorithm which probably deserves a more detailed discussion in the section 2.2.

**RE: Thanks for the advice. The whole paragraph is totally rewritten. New description of the sets of experiments is now on P6L9~15.**

The second sentence contradicts the first one since the first sentence mentions only "2 observational settings – one atmosphere (x2), and one ocean (w)" while the second sentence assumes assimilation of η observations "if applicable". The third sentence ignores assimilation of η again. The reader has to guess what are the "few PE cases with full SE" and why there are exactly 8 PE cases with partial SE. If assimilation of η observations is disregarded, with "2 observational settings" and "2 parameter settings" the reader will not be able to get 8 PE cases with partial SE.

**RE: The first sets of 8 experiments are trying to study the parameters on "air-sea" interaction. The later experiments of $\eta$ expand the work onto the deep ocean. New description of this difference is also added into the paragraph on P6L9~15.**

In my view, the authors should perform a significant work on "polishing" the manuscript to improve clarity of the material presentation.

**RE: The new revision is carefully checked and polished. Thanks.**

Comment 2. As with any suboptimal data assimilation method, the results of state and parameter estimation presented in the manuscript depend on particular realization of the method. That is why it is important to provide a clear description of the method what is sufficiently detailed to allow other researchers to repeat the calculations. I believe that the section 2.2 falls short of these expectations. It seems that at least the Reviewer 3 of the previous version of the manuscript did not get clear understanding of the method either. The Reviewer 3 wrote in his comments:

"Eq. (2) does not contain observation error statistics, and I am curious how to interpret this equation intuitively. I understand that this equation gives analysis increments for the ith ensemble member. The analysis increments should balance between the observation error and background error. This equation has only the background error variance in the observation space as the denominator, but does not contain the observation error variance which usually appears in the data assimilation equations as an R matrix".

In my view, the authors' reply to this comment is disappointing and the revisions to the section 2.2 did make the method description even vaguer. The authors' reply to the Reviewer 3 comment states:

"The observation error variance is calculated before this projection process. The observation are firstly compared to their simulated values, the difference between them are manipulated to produce the observational increments. The production of the observational increments considers the observation error variance and its PDF. The observation error is set as a constant number in our simulation. The standard deviation of "observational" errors are 2 for the atmospheric variables x1,2,3 and 0.2 for the oceanic variable w. New introduction of the EAKF method is added in the section 2.2, P4L1~10".

The following statements are at least inaccurate in this reply:

"The observation error variance is calculated before this projection process" – I believe that the data error variance is set, not computed, since the authors state: "The standard deviation of "observational" errors are 2 for the atmospheric variables x1,2,3 and 0.2 for the oceanic variable w".

"...The difference between" observations and their simulated values is "manipulated to produce the observational increments". What is this "manipulation"? There is no description of the procedure of this "manipulation" in the revised or original versions of the manuscript.

"The observation error is set as a constant number in our simulation" – is it really so? Isn't the observational error a random variable?

**RE: The vague description on observational error etc. is rewritten in section 2.3 (P6L2~L4, P6L15). Thanks for reviewer's thoroughly examination.**

The revised version in the section 2.2 is also quite confusing. P4L4 states:

The method "combines observational probability distribution function (PDF) with model PDF but under an adjustment idea. The algorithm can be sequentially implemented in a two-step procedure (Anderson, 2003): step 1 uses two Gaussian convolution to compute the observational increment at the observational location..."

First of all, two convolutions of what? Conventionally convolution is an operation on two functions resulting in another function. Do the authors mean the convolution of two Gaussian PDFs? But the observational increment is not a function, so it cannot result from convolution of two PDFs. Secondly, what is the "observational probability distribution function"? Do the authors mean PDF of data error? That is not the same as the PDF of observations. What is the "model PDF"? Model is not a random variable, it is an operator. Do the authors mean PDF of model forecast error? Do the authors know the PDF of the model forecast error to combine it with observational probability distribution function under an adjustment idea?

**RE: Description of the method is largely rewritten as in section 2.2. New equation (2) is added (P4L19) to explain how we calculate the observational increment before using them to perform the SE or PE by equation (3).**

Trying to find the answer to the question of the Reviewer 3 I came to the following interpretation of formula (2) in the manuscript: The authors seem to use well-known stochastic type of ensemble Kalman filter (see e.g. Hamil, 2006) where each ensemble member $x_i$ (model state and parameters) is updated to fit somewhat different realization of observations $y_i$ at the analysis time:

$\Delta x_i = K \Delta y_i$; $\Delta y_i = H(x_i) - y_i$, $i=1,...,m$ where $y_i = y + e_i$, and $e_i \sim N(0,R)$ is a realization of Gaussian random variable "observational error" with variance (covariance) R. Realizations $e_i$ are different for each ensemble member. The expression for the Kalman gain matrix K in formula (2) is an ensemble estimate for the case of a single observation of the conventional Kalman gain matrix given by the expression: $K = PH^T (HPH^T + R)^{-1}$ For example, $std(\Delta y)^2$ in the denominator in formula (2) approximates $HPH^T$-R: $std(\Delta y)^2 = <(\Delta y_i)^2> = < (H(x_i) - y_i)(H(x_i) - y_i)^T>$ , where $<>$ denotes ensemble averaging. Under the assumptions that the model is unbiased and e and x are uncorrelated, the expression above can be transformed to $<(Hx'_i - e_i)(Hx'_i - e_i)^T> = H<x'_i x'_i^T>H^T + <e_i e_i^T>$ while $<x'_i x'_i^T>$ and $<e_i e_i^T>$ tend to P and R respectively as the dimension of the ensemble increases.

If my guess is wrong and the the analysis for all ensemble members is performed using the same realization of observations, the data errors are effectively neglected by the filter given by equation (2).

If my guess regarding the meaning of the formula (2) is correct, the authors should describe the procedure of generation of the ensemble of observations in section 2.2. This is an essential part of the method (the method cannot use unperturbed observations). Unfortunately, the authors discuss the observations only in section 2.4 (P5L9):

"The output of last 103 TUs is then used as the "truth" to produce "observations" by superimposing a white noise on the "observed" variables. The standard deviation of "observational" errors are 2 for the atmospheric variables x1,2,3 and 0.2 for the oceanic variable w."

Note that this statement does not specify that the statistics of observational noise is Gaussian. The term "white noise" probably assumes no time correlations in observational errors which is always taken for granted in applications of the Kalman filter. Also it is not clear if the authors "produce" only one realization of observations by superimposing a white noise and "truth" or they produce an ensemble of perturbed observations.

In my view, the authors should revise the description of the data assimilation method considerably. The present description of the Ensemble Kalman filter method misses some important details which are critical for understanding the data assimilation results.

**RE: The discussion on the filtering algorithm with the realization of observation is added to the place that describes the method in section 2.2 (P4L13-21). Thanks.**

Comment 3. A number of inconsistencies can be found also in the description of the data assimilation results. For example, the authors mention several times that only the values of x2, w, and "(η if applicable)" are perturbed with noise and used as observations. The formula (2) assumes assimilation of scalar (single) data at the analysis time with separate data assimilation for "atmospheric" and "oceanic" sub-models. Conversely, in the statement:

P5L9 "The standard deviation of "observational" errors are 2 for the atmospheric variables x1,2,3 and 0.2 for the oceanic variable w."

the authors specify standard deviations for observational errors in x1 and x3 as if these variables were also used as observations (and do not provide standard deviation for η). Caption to Figure 1 also declares assimilation of x1,2,3 data: "a) both the atmosphere (x1,2,3) and ocean (w) from their observations (x1, 2, 3 and w)".

**RE: The observational error of $\eta$ is added on P6L15. In fully SE cases, $x_1$ and $x_3$ are constrained by the observation of $x_2$. These are clarified on P5L22~26. Thanks.**

Another inconsistency in the description of the results is related to the authors' discussion of the time scales of variability of "fast-varying variables of the atmosphere" (P3L13) and "the low-frequency variables of the ocean" (P3L13). The authors state that "the time scale of the w is nearly 10 times of the time scale of the x2" (P3L13), and that the "conceptual model mimics very fundamental natures of interactions of three typical time scales in the real world: synoptic (chaotic) atmosphere, seasonal interannual upper tropical oceans and decadal/multi- decadal deep ocean". This time scale separation was used by the authors as an argument for utilization of a simplified "weakly coupled data assimilation" approach (P5L20):

"x2 observations impact on all x variables, and w (η if applicable) observations impact on w (η) itself (considering the different time scales of w and η, no cross-impact between them)".The time scale separation is addressed several times in the interpretation of the results and is used as an explanation of the performance of different parameter estimation experiments, for example: "The energy of x2 is in the high frequency band and the energy of w is in the low frequency band. x2 varies fast and represents the most uncertain mode, transferrable to low frequency w through the "air-sea" interaction"(P8L19).

In the first reading of the manuscript, the description of the experiments and analysis of the results convinced me that the atmospheric variable x2 has a well determined spike of energy at periods of few TU, the variable w reacts to coupling with chaotic x2, but is mainly driven by the periodical forcing at the period of 10TU, while η has even longer scales of variability. The understanding that this impression is completely wrong came to me only when I read Appendix 2 there the authors describe an application of the high-pass filter to time series of η observations and states to remove "chaotic signal" and improve parameter estimation. The authors write:

P12L30 "... the periodic signal produced by the cosine function has a period of 10TUs (1000 time steps) (defined by Spd in Eq. (1), also see Fig. 10) and the chaotic signal is much slower than the periodic signal".

Since "chaotic signal" comes only from atmospheric variables, I was puzzled how could it be "slow". I had to read the results of the experiments with one-way coupled model presented in the section 3.2 several times to find this out. The turning point was Figure 10 showing time evolution of η in the experiments where the amplitude of the periodical forcing was increased 100 to 1000 times compared to the fully coupled model experiments. This figure clearly shows that influence of stochastic atmosphere on oceanic variables is not confined to the periods of few TUs, but dominates the whole range of resolved periods in time series of η. Chaotic atmospheric signal clearly larger than periodic signal in Figure 10 for amplitude of periodical forcing Ss=100, while in the fully coupled model Ss =1. Also, the authors mention that:

P10L14: "Comparing Fig. 9a to Fig. 6b, it can be seen that the chaotic signal in the one-way coupling model is much smaller than in the original two-way coupling model..."

My suppositions are that: a) for the experiments with fully coupled model (amplitude of the periodical force Ss=1) the ocean variables are driven almost entirely by the stochastic atmosphere while the periodical forcing in the equation for w seems to be negligible. b) the influence of stochastic atmosphere on oceanic variables is not confined to the periods of few TUs, but dominates the whole range of resolved periods in the ocean. c) time scale separation does not exist and cannot be used as a validation of a simplified "weakly coupled data assimilation" approach. In the light of these suppositions, the authors' conclusions that:

"enhancing estimation accuracy of atmospheric states is very important for the success of coupled model parameter estimation, especially for the parameters in the air-sea interaction processes" or P8L9: "This seems different from our previous intuition that in-situ ocean data are always considered as the first important piece of information for determining the oceanic coefficients. Our results here strongly suggest that in the future real coupled model PE experiments, for determining the best coefficient values, no matter the atmospheric or oceanic, sufficient and accurate atmospheric measurements are crucially important".

looks rather as the consequence of the particular settings of the "conceptual model" parameters and utilization of "weakly coupled data assimilation" approach, than as a general feature of parameter estimation that can be extrapolated to realistic climate models.

**RE: There is clear-cut on time scale separation for x and *w*. Comparing Fig. 6a to 6b, we can see both $x_2$ and *w* have the chaotic component through the whole period scope, but obviously *x* is more in high frequency and w is more in low frequency. For the two-way coupling cases, the chaotic atmospheric forcing is stronger than the periodical forcing in the system. The ocean feeds back to the atmosphere in the low-frequency band. The uncertainty caused by chaotic atmosphere exists in whole**

**range of resolved periods for both the atmosphere and the ocean. New discussions are added on P3L22~26 below the equation (1).**

Finally, note that the amplitude of the periodical force is stated to be 1 in the section 2.1: P3L18: "(c1, c3, c4, Sm, Ss, Spd, Γ, c5, c6) are

(10-1, 10-2, 10-2, 10, 1, 10, 102, 1, 10-3)" while in section 3.2 the authors state that Ss=10: P10L14: "Comparing Fig. 9a to Fig. 6b, it can be seen that the chaotic signal in the one-way coupling model is much smaller than in the original two-way coupling model (with an identical Ss value of 10)".

In my view, the authors should revise the description of the model and the sections presenting the results of data assimilation experiments considerably (and possibly redo the experiments to change the relative role of the periodical forcing). The present description may mislead the reader in the same way as it happened to me in my first reading of the manuscript.

**RE: Thanks for the advice. We rerun one-way coupling experiment of $s_s$=1 and the Fig. 9a is changed from $s_s$=10 to $s_s$=1. Effects of two-way to one-way coupling and varying $s_s$ experiments are separated. The referring paragraph is largely rewritten as in the place of P10L32 - P11L11.**

Minor remarks

1. Some terms used in the manuscript seem to be misleading and/or not well defined.

These terms are:

a) "signal-to-noise ratio of error covariance between the model state and parameter". The term signal- to-noise ratio is commonly introduced for time series of state variables, but not for covariances. I agree that it can be used for time series of covariances but that may mislead the reader. Why the authors do not use R shown in Figures 7 and 8 to analyze "relationship between the states and the parameters" quantitatively (I believe R in Figure 7 and 8 is signal-to- noise ratio", especially because it is not shown in the manuscript that rs2n is a better indicator of parameter estimation performance. Also, in more realistic applications, it might be impossible to assess the ratio Sf/Sp since the full set of observations is not available.

Separation of the signal and noise assumes that where is a model for noise or signal. Here this model is a linear relationship between state variable anomalies and parameter anomalies. I do not see why state variable anomalies and parameter anomalies should exhibit linear relationship at a particular time step. The authors' explanation of this linear relationship is a bit confusing:

P8L26: "PE completely relies on the covariance between the parameter and model states for projecting the observational information of states onto the parameter. While the PE projection is carried out by a linear regression equation based on the state-parameter covariance (EnKF/EAKF, for instance), only a linear or quasilinear R2= 1 -

||Hx-y||2/||y-mean(y)||2 which is conventionally called thecoefficient of determination of linear regression or Least Squares fit). It is possible to call R the time-mean "correlation coefficient". I do not see the reason to multiply R with Sf/Sp and call it the "relationship between parameters and states in ensemble is recognized.

**RE: The Pearson relation coefficient *R* is calculated with the following equation:**

$$R = \frac{1}{n}\sum_{i=1}^{n}\left(\frac{x_i - \bar{x}}{\text{std}(x)}\right)\left(\frac{y_i - \bar{y}}{\text{std}(y)}\right) = \frac{\text{cov}(x,y)}{\text{std}(x)\text{std}(y)}$$

$$\text{cov}(x,y) = \frac{1}{n}\sum_{i=1}^{n}(x_i - \bar{x})(y_i - \bar{y})$$

**And for the regression coefficient:**

**If we use *y=a+bx* to regress a linear relation between *x* and *y*, we can plot the points (*x*₁, *y*₁), …, (*x*ₙ, *y*ₙ) in the relation map to see how good is the linear relation. With the least square method:**

$$b = \frac{\sum_{i=1}^{n}y_i x_i - \left(\sum_{i=1}^{n}y_i\right)\left(\sum_{i=1}^{n}x_i\right)/n}{\sum_{i=1}^{n}x_i^2 - \left(\sum_{i=1}^{n}x_i\right)^2/n}$$

$$= \frac{\sum_{i=1}^{n}(y_i - \bar{y})(x_i - \bar{x})}{\sum_{i=1}^{n}(x_i - \bar{x})(x_i - \bar{x})}$$

$$= \frac{\text{cov}(x,y)}{\text{cov}(x,x)}$$

**It can be derived that:**

$$y = bx + a = \frac{\text{cov}(x,y)}{\text{cov}(x,x)}x + a = R\frac{\text{std}(y)}{\text{std}(x)}x + a$$

**Still, R is only a part of the signal-to-noise ratio, and the tilt of the line (R) is the same scale as the averaged distance of the fitting result to the "observation". Therefore Sf/Sp is especially important under a quasi-linear case. As mentioned by the reviewer, the Sf/Sp is hard to get in real world cases. Table 3 gives the results for R, Sf/Sp and rs2n as choice for real world case studies in the future.**

A hypothesis for all the failed cases without direct atmospheric SE could be that, under such a circumstance, the chaotic disturbances in the atmosphere (Lorenz equations in this case) continuously interacting with the parameter make difficulties for the system to build up a quasi-linear relationship between the state variable and the parameter."

I agree completely with the first sentence. The second sentence is misleading because the results of linear regression shown in Figures 7 and 8 do not correspond to formula (2). But I agree that "only a linear or quasilinear relationship between parameters and states in ensemble is recognized". I do not like the last sentence since no "interaction" between parameters and state is possible. Model states do not act on parameters. Instead of the last sentence I would suggest the following explanation:

PE completely relies on the covariance between the parameter and model states for projecting the observational information of states onto the parameter. The ensemble gets the information on parameter-state covariance entirely from dynamics (not from data): ensemble members obtained in the model forecast with different values of parameters become different with time. The ensemble of states accumulates this covariance information gradually, over time periods which are significantly longer than the data assimilation window of 0.05TUs. Partial assimilation of ocean data effectively restarts the model at every analysis step since oceanic variables are driven by x2 which is not controlled by observations (see Comment 3). As a result, almost no "dynamical" information is accumulated in ensemble covariance. Alternatively, in full data assimilation or in assimilation of x2 data, state corrections at the analysis step are more gentle and some dynamical information on state-parameter covariance has a chance to accumulate with time. And it is not that important if the relationship between parameters and states at a given time step is linear, non-linear or in general there is no relationship at all. It is important that ensemble estimates properly dynamically induced state-parameter covariance.

**RE: Thanks for the suggestion. The last sentence is removed in the revision and the new expression is on P9L19-21.**

b) "state-parameter covariance uncertainties"- Use of this term assumes that covariance is a random variable that has a "true", mean, or the most probable value. Then "uncertainties" are assessed as some measure of the higher moments of this random variable. This is not discussed in the manuscript.

**RE: The term of "state-parameter covariance uncertainties" is changed to "the inaccuracy of estimated state-parameter covariance" (P1L15-16).**

c)"chaotic-to-periodic ratio"- The term itself is fine, but no expression to compute this ratio is given. No numerical values of this ratio for different experiments are presented in the manuscript. It is only mentioned how the authors change this ratio:

P10L11: "Then we define a chaotic-to-periodic ratio (CPR) in the signals of w ($\eta$) by manipulating the coefficient Ss. Eight experiments are performed here, four for w-to-c2 PE and four for $\eta$-to-c6 PE. Each experiment has a different Ss value of 100, 250, 500 and 1000 and thus a reducing CPR in w and $\eta$".

It is worth to note that by changing the amplitude of Ss the authors also change RMS variability of w and $\eta$ but do not change variance of corresponding observational error. The observations may become much more "accurate" with increase of Ss by the factor of 1000.

Due to vague definition of signal-to-noise ratio and chaotic-to-periodic ratio the authors mix these two terms. In the Appendix 2 signal-to-noise ratio is used instead of chaotic-to-periodic ratio.

Also please correct the typo: P12L25 "signal-to-ratio ratio".

**RE: New definition of CPR is added in Appendix B. The CPR values of $w$ and $\eta$ are added on P11L6 and P11L9. Some signal-to-noise ratio is changed to CPR in appropriate places.**

3. P3 Eq(1): one of the coefficients Om, Gamma, or Od is redundant.

**RE: $O_d$ is damping coefficient. In this simple model, the damping coefficient is set identical for the upper ocean and deep ocean as 1. The parameter $O_m$ ($\Gamma$) combining with the damping coefficient $O_d$, defines the characteristic time scale. For example, the ratio $O_d/O_m$ of $10^{-1}$ ($O_d = 1$, $O_m = 10$) defines the characteristic time scale of $w$ being 10 times of that of $x_2$. New lines introducing these parameters and the value justification are added on P3L17-22.**

4. P4L26: "Thus a practical ensemble size of 20 (applicable for a CGCM) is chosen as a basic experiment setting". No comparison with CGCM is possible. In the presented research the ensemble size exceeds the dimensions of both the state vector and data. This is not the case for CGCM.

**RE: The justification with CGCM is removed. More appropriate justification is added on P8L4-7. Thanks.**

5. P4L30: "as 0.05 TU (i.e. 5 time steps)". Notation "TU" is not defined on page 4. It is introduced only on page 5.

**RE: Thanks for the suggestion. It is fixed in the revision (P5L18).**

6. P5L31: "The ensemble initial values of c2 are set as N(0.8, 0.5)". What if a particular realization of c2 is negative? The authors state that c2 is "analogous to the drag coefficient cd" which must be positive.

**RE: $c_2$ is restricted to be positive definite. New line is added on P6L22.**

7. P7L20: "We also performed the experiments under different update interval settings. Test results show that for the issue we are addressing, the conclusion is not sensitive to the update interval if it is within a reasonable range." The range of the tested update intervals should be given here.

**RE: New line is added on P8L9-10.**